# Single-cell lipidomics enabled by dual-polarity ionization and ion mobility-mass spectrometry imaging

Hua Zhang[1], Yuan Liu [1], Lauren Fields [2], Xudong Shi[3], Penghsuan Huang [2], Haiyan Lu[1], Andrew J. Schneider [4], Xindi Tang[2], Luigi Puglielli [4,5], Nathan V. Welham [3] & Lingjun Li [1,2,6,7] ✉

Single-cell (SC) analysis provides unique insight into individual cell dynamics and cell-to-cell heterogeneity. Here, we utilize trapped ion mobility separation coupled with dual-polarity ionization mass spectrometry imaging (MSI) to enable high-throughput in situ profiling of the SC lipidome. Multimodal SC imaging, in which dual-polarity-mode MSI is used to perform serial data acquisition runs on individual cells, significantly enhanced SC lipidome coverage. High-spatial resolution SC-MSI identifies both inter- and intracellular lipid heterogeneity; this heterogeneity is further explicated by Uniform Manifold Approximation and Projection and machine learning-driven classifications. We characterize SC lipidome alteration in response to stearoyl-CoA desaturase 1 inhibition and, additionally, identify cell-layer specific lipid distribution patterns in mouse cerebellar cortex. This integrated multimodal SC-MSI technology enables high-resolution spatial mapping of intercellular and cell-to-cell lipidome heterogeneity, SC lipidome remodeling induced by pharmacological intervention, and region-specific lipid diversity within tissue.

Characterization of molecular heterogeneity at the single-cell (SC) level is attracting unprecedented interest across the life sciences[1–4]. Large-scale SC genomic and transcriptomic profiles are routinely achieved via deep sequencing techniques[5–7]; however, unlike in SC genome sequencing, there are no tractable tools to amplify the proteins, peptides, and metabolites within individual cells, posing a substantial challenge to efforts in SC proteomics, metabolomics, and lipidomics[1,8,9]. This analytical challenge is compounded by the low abundance, large diversity, and dynamic physiology of these cellular components.

Mass spectrometry (MS) has recently emerged as a powerful tool for SC molecular characterization[1,4,10–13]. Current efforts are focused on harnessing the superior sensitivity of MS to enable SC interrogation with high information coverage and throughput and—in imaging applications—subcellular spatial resolution. This goal has spurred the development of new MS strategies for SC analysis. One strategy, coupling SC micro-sampling with direct nano electrospray ionization (nanoESI)-MS[14–16], has the capability to detect circa 100 metabolites per cell; however, this approach is labor intensive as it requires precise manual operation during SC micro-sampling and direct nanoESI-MS analysis, resulting in limited throughput and poor reproducibility. Other strategies, such as flow cytometry coupled with nanoESI-MS[17,18] or inductively coupled plasma (ICP)-MS[19–21], have yielded substantial improvements in throughput but offer limited molecular coverage;

[1]School of Pharmacy, University of Wisconsin-Madison, Madison, Wisconsin 53705, USA. [2]Department of Chemistry, University of Wisconsin-Madison, Madison, Wisconsin 53706, USA. [3]Division of Otolaryngology, Department of Surgery, School of Medicine and Public Health, University of Wisconsin-Madison, Madison, Wisconsin 53792, USA. [4]Department of Medicine, University of Wisconsin-Madison, Madison, WI 53705, USA. [5]Waisman Center, University of Wisconsin-Madison, Madison, WI 53705, USA. [6]Lachman Institute for Pharmaceutical Development, School of Pharmacy, University of Wisconsin-Madison, Madison, WI 53705, USA. [7]Wisconsin Center for NanoBioSystems, School of Pharmacy, University of Wisconsin-Madison, Madison, WI 53705, USA. ✉e-mail: lingjun.li@wisc.edu

such MS cytometry approaches also forgo subcellular spatial localization information when injecting the whole cell for analysis.

Laser/ion-beam based methods, such as secondary ion mass spectrometry (SIMS)[22–25], laser desorption ionization (LDI)[26,27], laser ablation electrospray ionization (LAESI)[28,29], and matrix-assisted laser desorption/ionization (MALDI)[30–37], offer an alternative and compelling approach to SC molecular characterization. For instance, elemental distribution within cells has been depicted by SC imaging with LID-MS[26,27]. Notably, MALDI enables soft ionization of endogenous cellular metabolites and lipids that are indispensable to biological understanding. High-throughput SC analysis via MALDI-MS imaging (MSI) has recently been demonstrated using methods such as SpaceM[30] and microMS[31,32]; however the lateral resolution (≥50 μm) achieved using these methods remains insufficient to interrogate subcellular features within individual cells. Given that the spatial resolution of MALDI-MSI has advanced to the submicron level as a result of instrumentation advances, including atmospheric pressure MALDI[36], transmission-mode MALDI, and laser post-ionization (t-MALDI-2)[33], subcellular resolution MSI is now considered methodologically feasible. For instance, with a novel optics system, atmospheric pressure MALDI[36] was able to achieve a lateral resolution of 1.4 μm, which could be further reduced to 600 nm for brain tissue via transmission-mode MALDI and laser post-ionization (t-MALDI-2)[33]. The lack of a molecular separation procedure following metabolite desorption/ionization presents an additional challenge to current SC MALDI-MSI approaches due to a considerable ion suppression effect. This challenge is particularly relevant to SC lipidomics, as lipids exhibit high structural complexity resulting from the presence of numerous isobaric/isomeric species[38,39]. There is still a strong demand for higher molecular specificity in SC MALDI-MS imaging.

In this study, we implement trapped ion mobility separation combined with dual-polarity ionization MSI to enable high-throughput in situ profiling of SC lipidomes. We first evaluate the performance of this approach using MSI analysis of SC samples from mono- and co-cultured human pancreatic cancer (PANC-1) and activated pancreatic stellate cells (PSC), as well as neuroblastoma cells (SK-N-SH); analyses are performed using a MALDI trapped ion mobility time-of-flight/time-of-flight MS (MALDI-timsTOF-MS) platform. Our findings demonstrate that high-throughput in situ SC mapping can be achieved, yielding rich molecular information in both positive and negative ion modes. We further characterize SC lipidome alteration induced by stearoyl-CoA desaturase 1 (SCD1) inhibitor treatment and elucidate cell-to-cell lipidome heterogeneity via Uniform Manifold Approximation and Projection (UMAP) and machine learning algorithms. Finally, we apply our SC-MSI platform to in situ profiling of lipid diversity in the mouse brain.

## Results

### SC-MSI at high spatial resolution

In high-throughput SC-MALDI MSI, key challenges include insufficient sensitivity for the extremely minute chemical components from an individual cell, as well as chemical diffusion when coating the MALDI matrix on the SCs. The schematic workflow of SC-MALDI-MSI is shown in Fig. 1. To develop a robust SC-MSI platform, the MALDI matrix and the matrix application method were first optimized (Supplementary Figs. 1–5, detailed in the "Methods"). Interestingly, as shown in Supplementary Fig. 5, the MALDI matrix-coated SCs could be clearly observed under the microscope after the sublimation, which allows for in situ MALDI spotting the SCs on the ITO slide, whereas the cells could hardly be recognized with matrix deposited via spray due to the large size of the matrix crystal. To further evaluate the reproducibility and accuracy of MALDI-MSI for SC analysis, lipid standards of LPC 18:1 and PC (18:1-18:1) were deposited onto a matrix-coated slide via an automated sprayer, and 12 square regions (ca. 200 × 200 μm, 400 pixels of each) from the slide were selected for MS imaging. Among the 12 regions, relative standard deviations (RSDs) for LPC 18:1 ([M + H]+, $m/z$ 522.3545) were 9.5–10.5%, and the RSDs were 11.8–14.3% for PC (18:1-18:1) ([M + H]+, $m/z$ 786.6007), while the average intensities of LPC 18:1 and PC (18:1-18:1) across the 12 regions were quite consistent, with RSDs ($n = 12$) of 1.35 and 1.67% for LPC 18:1 and PC (18:1-18:1), respectively (Supplementary Fig. 6a, b). Also, the average mass spectra from the 12 regions showed that the peaks of LPC 18:1 and PC (18:1-18:1) were in a good alignment (Supplementary Fig. 6c, d). The results indicate that the high-resolution MALDI-MSI could offer good reproducibility and mass accuracy for SC analysis. Indeed, signal intensity variations were found within each region; this could be due to incomplete homogenous deposition of the lipid standards onto the slide surface by using a pneumatic sprayer, and the lipid solution microdroplets could also affect the MALDI matrix crystal upon its landing on the surface of the slide.

In situ SC spotting showed that abundant lipid species were obtained from the SC via sublimation (Fig. 2a). MSI images of SC lipids with clear lateral resolution were obtained owing to the reduced chemical diffusion (Fig. 2). In comparison with the microscopic image

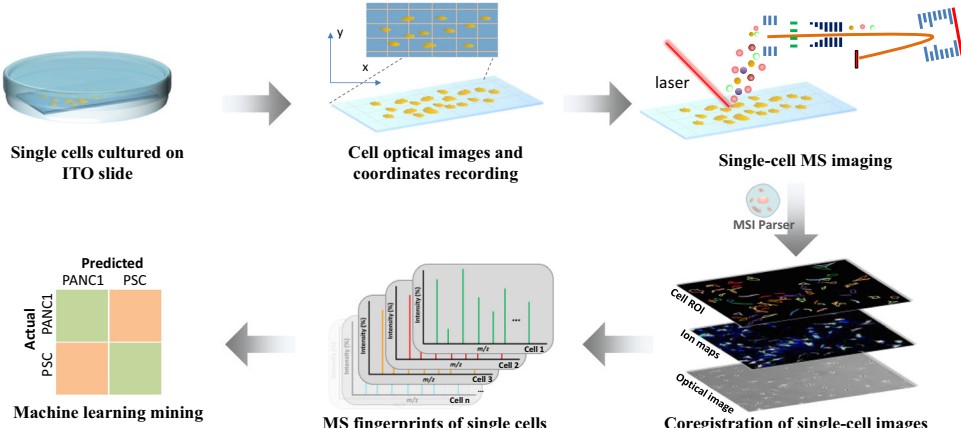

**Fig. 1 | Schematic workflow of single-cell (SC) analysis based on high-spatial-resolution MALDI-MS imaging.** The individual SCs were cultured on an ITO slide, where precise cellular coordinates were established using a microscope. Subsequently, the SCs underwent a MALDI matrix coating process through sublimation. The SC slide was then subjected to MALDI-MSI, which involved the integration of ion mobility separation and dual-polarity ionization techniques. To gain a deeper understanding of cell-to-cell lipidome heterogeneity, a custom-developed MSI Parser program was developed to pick the MS data from each individual cell. This program harnessed the power of UMAP and machine learning techniques to unravel intricate lipidomic variations among cells. For details about the experimental workflow refer to the Method section.

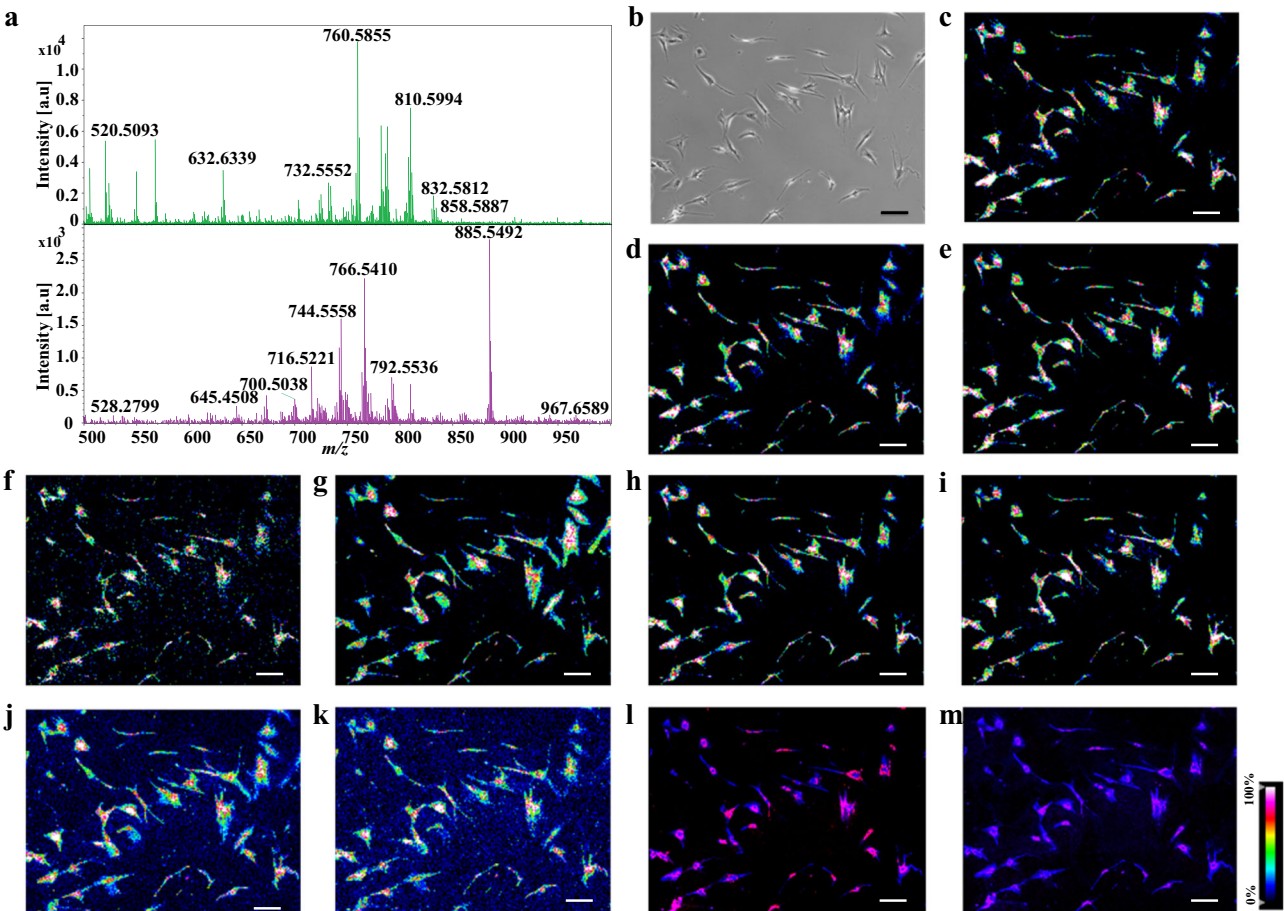

**Fig. 2 | SC MS imaging of the PSC cells. a** Mass spectra obtained from the PSC SC sample under positive-ion detection mode (the top panel) and negative ion detection mode (the bottom panel), **b** Bright-field image of the PSC cells prior to the SC-MSI, the data are representative of three independent experiments. MS images of representative lipid species detected from PSC cells, **c** PC (32:1) ([M + H]⁺, $m/z$ 732.5537), **d** PC (34:2) ([M + H]⁺, $m/z$ 758.5685), **e** PC (34:1) ([M + H]⁺, $m/z$ 760.5854), **f** PC (O-36:2) ([M + H]⁺, $m/z$ 772.5846), **g** PC (36:4) ([M + H]⁺, $m/z$

782.5685), **h** PC (36:2) ([M + H]⁺, $m/z$ 786.5933), **i** PC (36:1) ([M + H]⁺, $m/z$ 788.6135), **j** PC (38:4) ([M + H]⁺, $m/z$ 810.6052), **k** PC (38:3) ([M + H]⁺, $m/z$ 812.6139), **l** overlay of Cer (34:2;O) ([M + H-H₂O]⁺, $m/z$ 520.5044, red) and PC (34:1) ([M + H]⁺, $m/z$ 760.5854, blue) ion images, **m** overlay of Cer (42:3;O) ([M + H]⁺, $m/z$ 630.6186, purple) and PC (34:1) ([M + H]⁺, $m/z$ 760.5854, blue) ion images. All SC-MSI images were obtained with a mass error tolerance of 10 ppm. Scale bar, 200 μm.

(Fig. 2b), the morphology of the SCs was precisely depicted via SC-MSI (Fig. 2b–k). Furthermore, SC-MSI of PANC-1 and SK-N-SH cells were also achieved (Supplementary Figs. 7 and 8). Note that lipid distribution within the SCs could even be revealed at near-subcellular resolution (Fig. 2l, m and Supplementary Fig. 9). For instance, the higher signal intensity of the ceramides was present in the perinuclear region, whereas it showed low signal intensity in the nuclear region. We also noticed that more lipid species could be revealed from SC with the assistance of ion mobility separation, as illustrated in Fig. 3, while only one peak was observed when using MS alone, the incorporation of ion mobility separation enabled the resolution and recognition of four distinct lipid species. In addition, variations in lipid isobaric/isomeric composition between the PSC and PANC-1 cells were observed with ion mobility separation (Supplementary Figs. 10 and 11). These results demonstrate that ion mobility provides a distinct advantage in revealing isobaric and isomeric lipid species in SC-MSI.

In addition to positive-ion mode SC-MSI, negative ionization mode with MALDI matrix of 1,5-diaminonaphthalene (DAN) was employed, and distinct lipid species not previously identified during positive ionization mode were detected (Fig. 2a, the bottom panel). Under the positive mode, the predominant lipid species detected from SCs were phosphatidylcholines (PCs), whereas lipid species, including fatty acids (FAs), phosphatidylethanolamines (PEs), phosphatidylserines (PSs), and phosphatidylinositols (PIs) were more readily detected

in negative mode. Supplementary Fig. 12 shows representative MS images of FAs and lysophosphatidylethanolamine (LPE) detected from the PSC cells under negative ionization mode. The negative mode analysis provides a substantial complement to the positive mode analysis in which spatial distributions of a total of 185 lipids were revealed from the SC samples, enabling higher molecular coverage of SC lipidome in comparison with previous studies[30,32,35]. It is noted that the identities of lipids in SCs were assigned based on tandem MS by in situ SC-MALDI MS spotting (Supplementary Fig. 13, and Supplementary Data 1 and 2) and HPLC-NanoESI-MS/MS analysis of cellular lipid extracts (Supplementary Data 4). It is worth mentioning that a data acquisition rate of ca. 9 mm²/h at 26 pixels per second could be achieved under the condition of 50 shots per pixel and a laser firing frequency of 10 kHz, therefore, allowing high-throughput profiling of SCs.

We noticed that the cells remained on the ITO slide with almost intact morphology after the SC-MSI experiment under positive mode (Supplementary Fig. 14), which might be attributed to the low laser ablation energy during SC-MSI. This inspired us to perform a second MSI run on the same SCs while using negative mode. After positive mode SC-MSI, the CHCA matrix residuals on the cell surface were washed away using 50 mM ammonium acetate aqueous, followed by coating with DAN matrix. The subsequent negative mode MSI enables abundant negative-ionization-prone lipid species such as

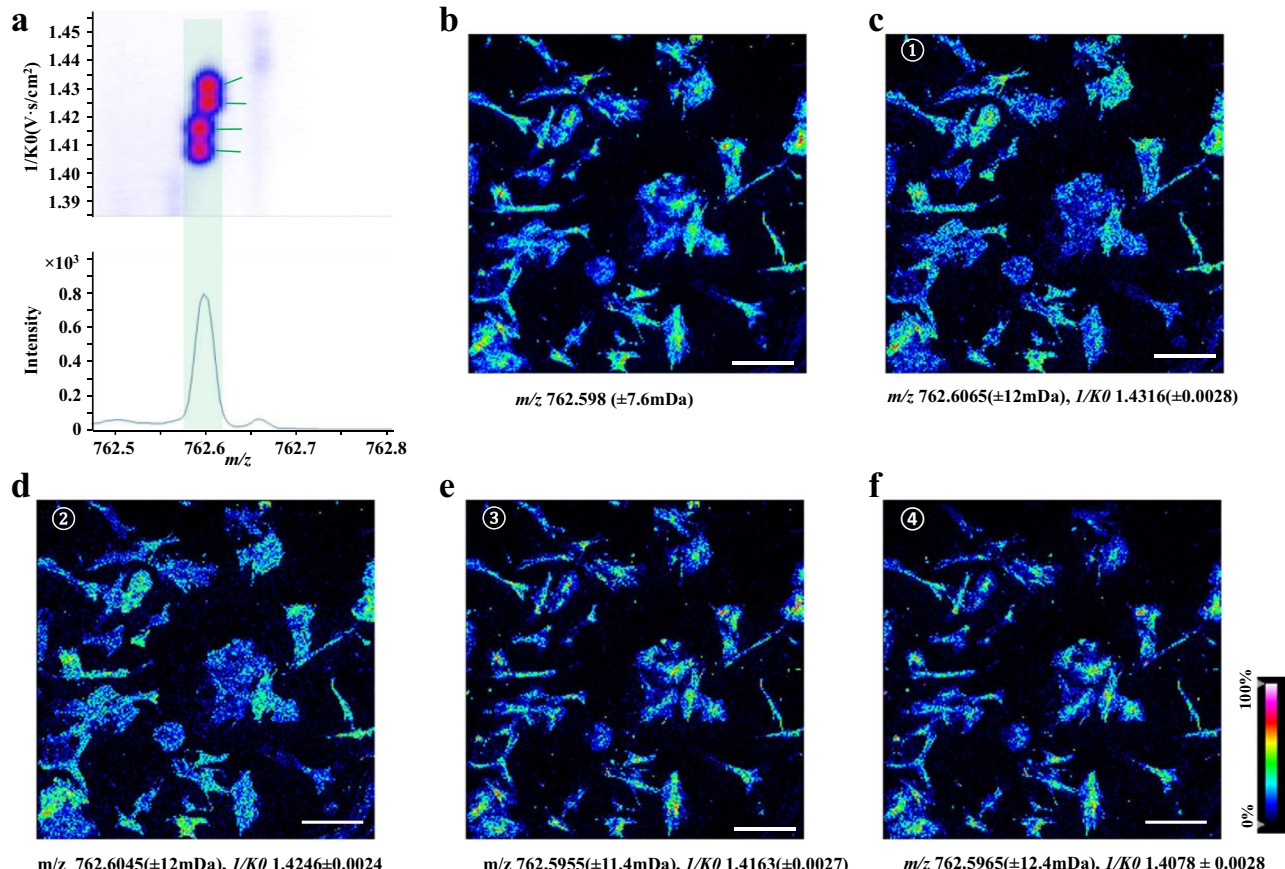

**Fig. 3 | SC-MSI of the PSC cells coupled with the ion mobility separation.**
**a** Representative ion-mobility MS incorporated data at a mass window of *m/z* 762.5 to 762.8 Da, **b** MS images constructed based on *m/z* value alone, **c–f** MS images of isobaric and isomeric lipids revealed based on incorporation of the *m/z* value and collisional cross-section (CCS) value information. Scale bar, 400 μm.

phosphatidylethanolamine (PE), phosphatidylserine (PS), and phosphatidylinositol (PI) to be detected from the same individual cells (Supplementary Fig. 15). The following negative mode MSI showed the distribution of lipids such as PE (34:1) at *m/z* 744.5552, PS (36:1) at *m/z* 788.5429, and PI (38:4) at *m/z* 885.5477 from the same individual cells (Fig. 4). As shown in Supplementary Fig. 16, the mass spectra exhibit a high degree of reproducibility as reflected from the consistency between the single-polarity ionized and dual-polarity ionized PSC cells, though the signal intensities of some lipid species were slightly reduced in the dual-polarity ionization scenario. Furthermore, the MS images obtained from the second MSI suggested that the delocalization of the lipid species was relatively minimal (Supplementary Fig. 17 and Fig. 4). As a result, multimodal MS imaging of individual SCs was achieved that further expanded the SC lipidome coverage.

## Cell-to-cell heterogeneity revealed via SC-MSI
With the SC-MSI platform, SC chemical heterogeneity within the same cell population, as well as different cell subpopulations, could be visualized and elucidated. While enormous efforts have been devoted to uncovering the SC chemical heterogeneity across different cell lines[40,41], recent research has revealed that substantial heterogeneity also exists within the same cell population, demonstrating significant variability from cell to cell[4,30,35]. The PSC cell MSI results showed varied lipid abundance among different individual PSC cells (Fig. 2), which might be related to the lipidome heterogeneity within the same cell population. For instance, some PSC cells exhibited higher signal intensity of PC (36:4) (*m/z* 782. 5685) than others (Fig. 2g). To further characterize the lipidome heterogeneity within the PSC cell population, mass spectra from each individual cell were extracted and

subjected to UMAP analysis. The results showed that the cells were categorized into subgroups (Supplementary Fig. 18), which confirmed the inter-population SC heterogeneity and highly dynamic nature of the SC lipidome. Theoretically, each individual cell might be in a different growth state and with distinct metabolic activity temporally, which is responsible for the SC lipidomic heterogeneity within the same cell line. It is worth noting that the analysis of SC heterogeneity can incorporate both positive and negative ion mode mass spectra (Supplementary Fig. 19), owing to the dual-polarity ionization technique employed in the analysis of a single cell. This dual-polarity ionization enables a more comprehensive assessment of the molecular composition and diversity within individual cells. Furthermore, as shown in Supplementary Fig. 20, the integration of ion mobility separation adds an additional dimension of molecular information to the profiling of SC heterogeneity.

Furthermore, PSC and PANC-1 cells were co-cultured to study the lipidome heterogeneity in different human cell subpopulations. These two cell types are from different lineages and have different morphology (Supplementary Fig. 21). Co-culture of PSC and PANC-1 cells is an essential in vitro model for the study of pancreatic ductal adenocarcinoma (PDAC), such as evaluating cancer progression and drug resistance, as well as studying the interactions between pancreatic tumor and stromal cells[40,41]. In situ SC MALDI spotting revealed distinct mass spectrometric fingerprints from the two types of cells (Supplementary Fig. 22). The SC-MSI of the co-culture cell sample is shown in Fig. 5. The results show that lipid species such as PC (32:1) at *m/z* 732.5337, PC (34:1) at *m/z* 760.5840, PC (36:4) at *m/z* 782.5685 are of high signal intensities in both cell lines (Fig. 5b, c, f), whereas lipids of PC (32:0) at *m/z* 734.5676 and PE (36:0) at *m/z* 748.5827 are of higher

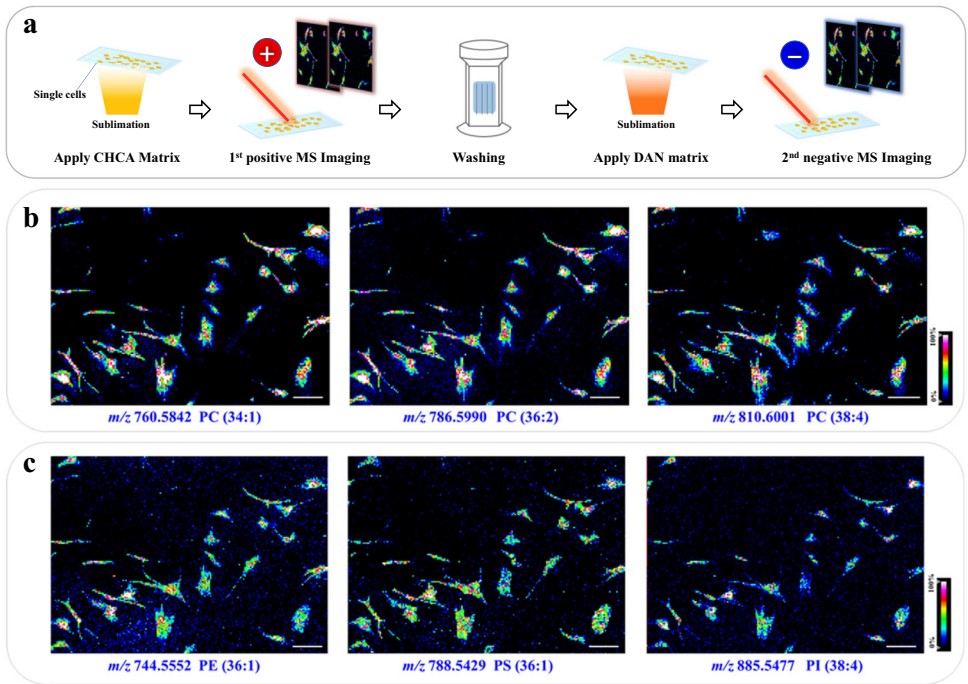

**Fig. 4 | Multimodal SC-MSI of individual cells. a** Schematic diagram of the workflow for multimodal SC-MSI of the single cells on the ITO slide, **b** MS images of phosphatidylcholines (PCs) from PSC cells under positive mode, **c** representative MS images of PE, PS, and PI acquired from the same PSC cells in a subsequent acquisition via negative mode SC-MSI. Scale bar is 200 μm, and the mass error tolerance is 10 ppm for each MS image.

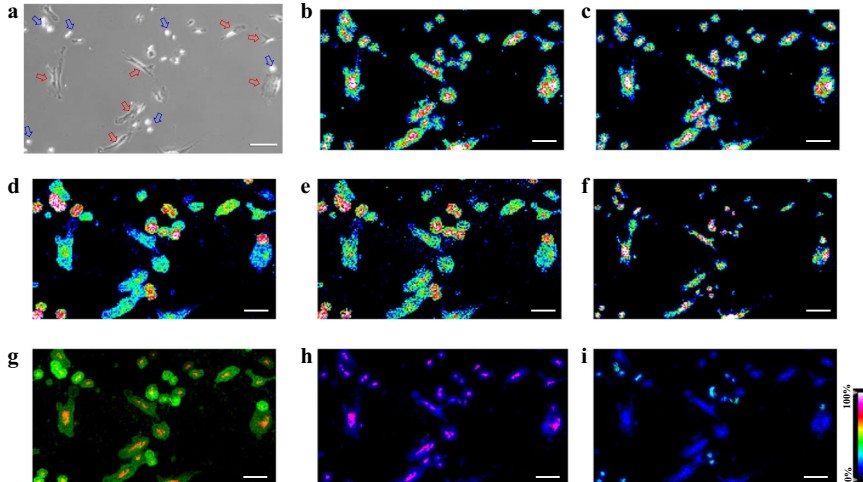

**Fig. 5 | SC-MSI images of PANC-1 and PSC cell co-culture samples. a** Bright-field microscopy images of the PANC-1 and PSC cell co-culture sample (blue and red arrows denote the PANC-1 and PSC cells, respectively), the data are representative of three independent experiments. MS images of representative lipid species detected from the cell co-culture, **b** PC (32:1) ([M + H]⁺, *m/z* 732.5537), **c** PC (34:1) ([M + H]⁺, *m/z* 760.5840), **d** PC (32:0) ([M + H]⁺, *m/z* 734.5676), **e** PE (36:0) ([M + H]⁺, *m/z* 748.5827), **f** PC (36:4) ([M + H]⁺, *m/z* 782.5685), **g** overlay of Cer (40:1;2 O) ([M + H-H₂O]⁺, *m/z* 604.6003, red) and PC (32:0) ([M + H]⁺, *m/z* 734.5676, green) ion images, **h** overlay of Cer (40:1;2O) ([M + H-H₂O]⁺, *m/z* 604.6003, red) and PC (34:1) ([M + H]⁺, *m/z* 760.5854, blue) ion images, **i** overlay of PC (34:1) ([M + H]⁺, *m/z* 760.5840, blue) and PC (O-36:4) ([M + H]⁺, *m/z* 768.5853, green). All SC-MSI images were obtained with a mass error tolerance of 10 ppm. Scale bar, 200 μm.

signal intensities in the PANC-1 cells (Fig. 5d, e). In addition, some lipid species were found to be highly cell-type dependent. For instance, the PC (O-36:4) at *m/z* 768.5853 was exclusively observed in the PANC-1 cells (Fig. 5i). As shown in Fig. 6a, two isobaric lipids were detected with different collision cross-section (CCS) values, in combination with the ion-mobility information, dramatically different distributions of these two isobaric lipids were found in the pancreatic cell subpopulations. UMAP analysis of the mass spectrometric fingerprints showed that the two pancreatic cell subpopulations were successfully classified into separate clusters (Fig. 6b). The Volcano plot highlights the lipid species with particularly pronounced variations between the PANC-1 and PSC cells (Supplementary Fig. 23a, Supplementary Data 3). For robust phenotyping of SCs, the lipidomic features of PANC-1 and PSC cells were subjected to machine learning-driven classification with models including support vector machine (SVM), random forest classifier (RF), and multilayer perceptron (MLP). The cross-validation results obtained via three algorithms show that all three models have 100% accuracy and 100% specificity and sensitivity in the classification and

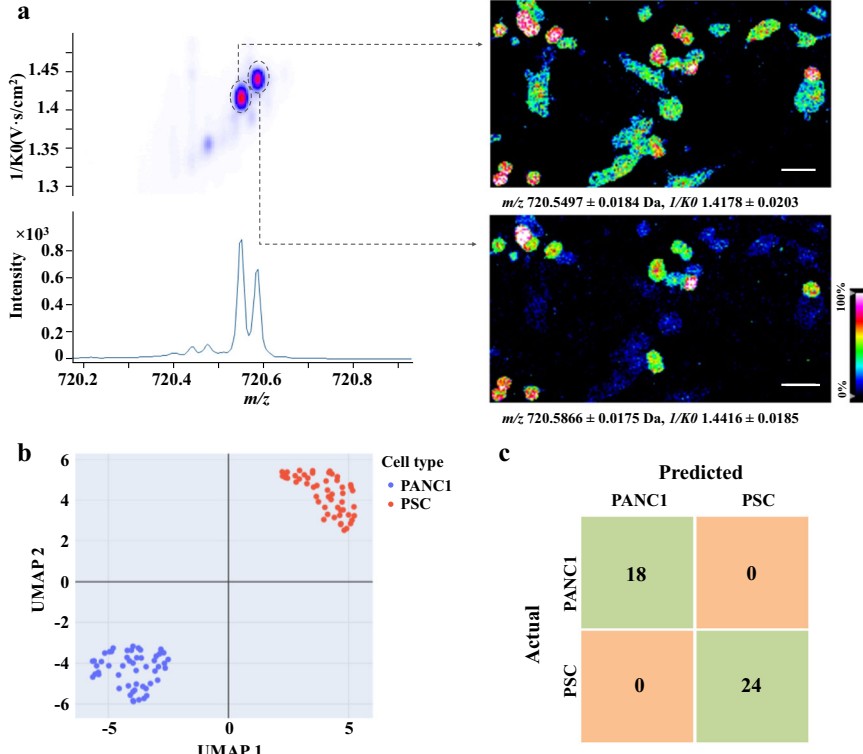

**Fig. 6 | SC heterogeneity revealed by SC-MSI analysis of PSC and PANC-1 co-culture. a** Ion-mobility MSI data collected for isobaric lipids at a narrow mass range of $m/z$ 720.2 to 720.8 Da (right panel) and the spatial distribution of the two isobaric lipids of PE (34:0) ([M + H]$^+$, $m/z$ 720.5497, 1/K0 1.4178) and PC (O-32:0) ([M + H]$^+$, $m/z$ 720.5866, 1/K0 1.4416) based on the combination of their $m/z$ value and the collisional cross-sectional (CCS) information (left panel), **b** UMAP analysis of the PANC-1 and PSC cell subpopulation, 52 SCs from each cell type, **c** The confusion matrix displays the results of the testing data using the support vector machine (SVM) model for the mass spectrometric fingerprint analysis of PSC and PANC-1 cells. Scale bar, 200 µm.

prediction of the cell type based on lipidome signature (Fig. 6c and Supplementary Fig. 23b, Supplementary Table 1).

### SC lipidome alteration in response to drug treatment

Studying the functional and compositional heterogeneity caused by drug treatment at the SC level remains challenging in drug discovery and development[42,43]. In our proof-of-principle study, the PSC cells were treated with MF-438, a stearoyl-CoA desaturase 1 (SCD1) inhibitor which inhibits the conversion of saturated fatty acid (SFA) to mono-unsaturated fatty acid (MUFA)[44]. The cell viability assay indicated that the MF-438 drug has significant effects on the PSC cell proliferation with concentration over 10 nM (Fig. 7a). The SC-MSI results show that higher signal intensities of the FAs were found in control (Fig. 7). For instance, significant downregulation of FA (16:1), FA (18:1), FA (20:1), FA (22:1), and LPE (18:1) was observed from the drug-treated group compared to the control. Interestingly, the signal intensities of FAs exhibited a more pronounced reduction compared to that of the LPEs. Pathway analysis indicates the downregulation of the FAs is closely related to the pathways of biosynthesis of unsaturated fatty acids (p-value, 4.34e-8) and linoleic acid metabolism (*p*-value, 0.035) (Fig. 7l), which is consistent with previous research showing that the inhibition of the SCD1 could block the synthesis of the MUFA[45,46].

### Discussion

SC analysis provides transformative insight into the complexity and diversity that are inherent to real-world biology. Within this domain, MS-based methodologies comprise a powerful toolkit for the study of SC omics at the molecular level; however, the full realization of the potential of SC proteomics, metabolomics, and lipidomics requires further innovation and improvements in sample preparation, instrumentation, and data acquisition. In this study, we employed trapped

ion mobility separation coupled with dual-polarity ionization in high-spatial resolution MALDI-MSI to enable high-throughput profiling of the SC lipidome. It is noted that subcellular resolution single-cell MS imaging has the potential to become a powerful tool for elucidating the roles of bioactive molecules within individual cells, offering new insights into intracellular biological interactions, and revealing previously inaccessible information about cellular heterogeneity. Figure 2 and Supplementary Fig. 9 suggest that ceramides were present in the perinuclear regions, indicating that near-subcellular resolution MSI is achieved with these single-cell analyses. Indeed, highly sensitive and high-spatial-resolution MSI techniques are still in demand for achieving subcellular resolution MSI. Recent parallel efforts to produce an advanced laser source with a smaller pixel size suggest that even subcellular resolution MSI is technically possible[33,35,36]. It is worth mentioning that the variation in height or depth of individual cells across their surface poses a significant analytical challenge for SC-MS imaging. SC-MSI would be enhanced by improved spatial resolution and a SC sectioning strategy to reduce the impact of potential variations resulting from changes in cell morphology.

Our results indicate that the washing and fixing pretreatments applied prior to SC-MSI have minimal impact on the lipidome of the cells on the slide (Supplementary Figs. 24 and 25). Indeed, it is worth mentioning that caution is required when fixing the single-cell sample with the formaldehyde PBS solution. The fixing time, temperature, and concentration of formaldehyde should be kept at a low level to avoid excessive fixation, as the overfixing can negatively affect the detection of amine-containing lipids such as phosphatidylethanolamine (PE), phosphatidylserine (PS).

Furthermore, given the varied charge properties inherent to different lipid classes and the consequent importance of both positive and negative ionization mode analysis[39], we reasoned that dual-polarity

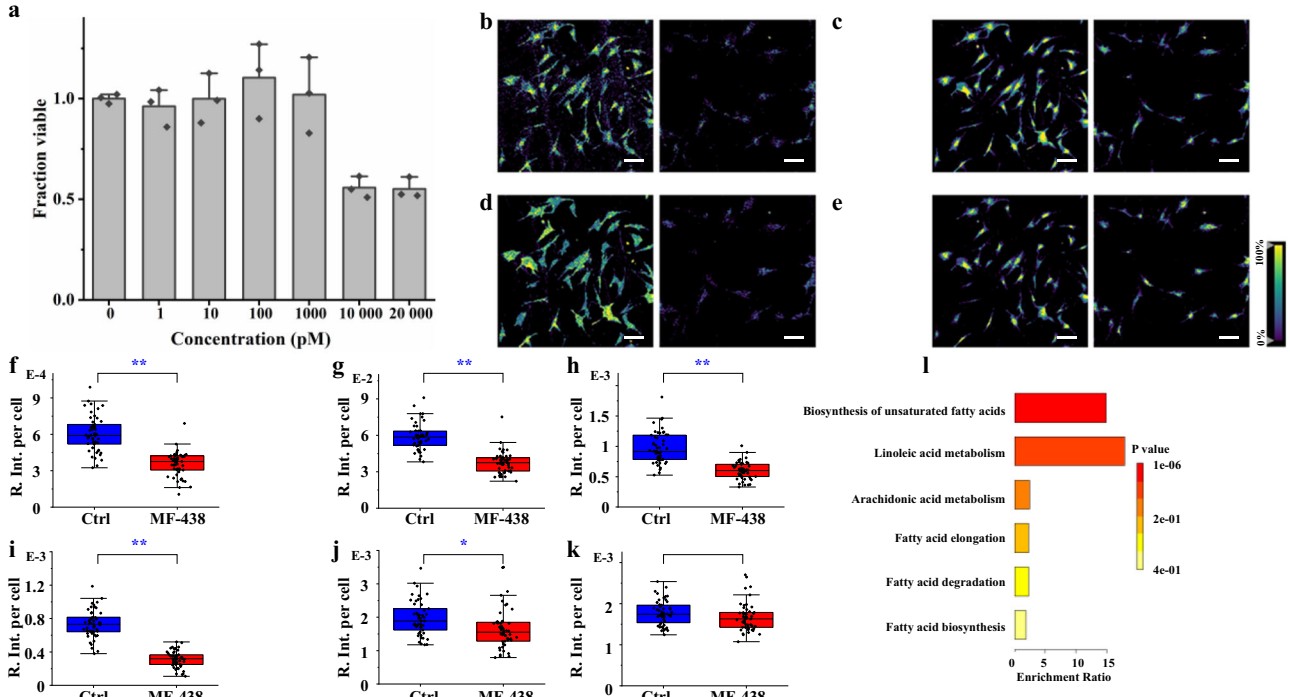

**Fig. 7 | SC-MSI analysis of PSC cells under MF-438 drug treatment. a** MF-438 dose-response on PSC cells. Three biological replicates are shown; representative MSI images of PSC SCs under control (left panel) and MF-438 drug treatment (right panel), black dots indicate replicate data points ($n$ = 3); error bars denote S.D., **b** FA (16:0) ([M-H]⁻, $m/z$ 255.2167), **c** FA (18:1) ([M-H]⁻, $m/z$ 281.2476), **d** FA (18:0) ([M-H]⁻, $m/z$ 283.2657), **e** FA (20:3) ([M-H]⁻, $m/z$ 305.2434), all MSI images were obtained with a mass error tolerance of 10 ppm. Scale bar, 200 μm; box plots of lipid species detected in the PSC SCs under the control and drug treatment conditions, **f** FA

(16:1) ([M-H]⁻, $m/z$ 253.2143), **g** FA (18:1) ([M-H]⁻, $m/z$ 281.2476), **h** FA (20:1) ([M-H]⁻, $m/z$ 309.2795), **i** FA (22:1) ([M-H]⁻, $m/z$ 337.3074), **j** LPE (18:1) ([M-H]⁻, $m/z$ 478.2967), and **k** LPE (20:1) ([M-H]⁻, $m/z$ 506.3261), the box plots were based on the relative intensity ratios (R. Int. pre cell) of each ions to the base peak of ($m/z$ 313.1428, derived from DAN matrix) from each individual cell, $n$ = 50, significant difference was determined by a two-tailed Student's $t$-test (*$p < 0.05$, **$p < 0.001$); **l** Pathway analysis of the downregulated FAs. All box plots indicate median (center line), 25th and 75th percentiles (bounds of box), and minimum and maximum (whiskers).

ionization MSI would yield improved SC lipidome coverage. This is consistent with previous studies[47–49], which showed that substantially enhanced molecular coverage is achieved via dual-polarity ionization MS imaging on the same tissue section. It is noted that SC-MSI also requires high mass spectral resolution to avoid the isobaric overlaps among different lipid species and careful consideration when constructing the MS images of SC (Supplementary Fig. 26).

Our work represents the multimodal MSI of an individual cell (Fig. 4). This approach offers an avenue for in-depth profiling of SC heterogeneity. Given that the cells in our experiments were identified with grossly intact morphology following data acquisition (Supplementary Fig. 14), we anticipate that a substantial amount of cellular material remained following lipidomic analysis, suggesting that other SC methodologies could be implemented in series, facilitating the acquisition of orthogonal and complementary datasets from each cell. In our experiment, the CHCA matrix was washed away with 50 mM ammonium acetate after the first MSI run. The extent of lipids delocalization caused by ammonium acetate washing is generally considered to be minimal. This has been confirmed by an earlier study that the use of ammonium acetate to wash mouse brain tissue sections can result in a noted improvement in signal intensity and lipidome coverage during negative ion mode MALDI-MS analysis without any indication of delocalization at a lateral resolution of 10 μm[50].

It is also worth mentioning that although trapped ion mobility aids the separation and recognition of lipid isomeric/isobaric species within SCs to some extent[51], this strategy may be inadequate to distinguish structural isomers, such as the C = C positional lipid isomers. Rather, chemical derivatization, such as via epoxidation[52,53], Paternò–Büchi reaction[54,55], or ozonolysis[56,57], coupled with tandem MS analysis, could facilitate structural lipidomic interrogation. We hypothesize that in situ

single-cell chemical derivatization could offer a promising approach for exploring structural lipidome in SC-MSI analysis.

Indeed, recent advances have shed light on important aspects of SC-MS analysis and have significantly advanced our understanding of the underlying molecular heterogeneity in SC[4,9,13,14,35]. While our study shares a similar sample preparation procedure in SC-MS imaging in comparison with previous publications[30,35], it has several distinctive features. First, a dual-ionization strategy for SC-MS imaging on the same individual cells has been developed, which significantly enhanced the lipidome coverage from a single cell. Second, incorporating ion mobility separation in SC-MSI provides a distinct advantage in revealing isobaric and isomeric lipid species. Moreover, a graphical user interface (GUI) platform was introduced for single-cell MSI data analysis, offering streamlined and automated single-cell MSI data analysis. These advances demonstrate the potential of our approach to provide valuable insights into the lipid heterogeneity of single cells.

We noted that for SC-MSI of complex tissue samples, the establishment of a precise correlation between the MS imaging results and each individual cell present in the tissue remains a significant challenge. This will be a crucial area in our future research. As an example, the high-spatial-resolution MSI profiles of mouse brain showed region-specific, cellular layers lipidome heterogeneity within the cerebellar cortex (Supplementary Figs. 27–29). It is also worth mentioning that a direct connection between the lipid signal intensities and its abundance from various regions may not be strictly feasible, as several factors can influence the lipid signal intensities in MALDI-MS, such as ion suppression effect and local salt centration[58]. Thus, to ensure more accurate quantification analysis, it is necessary to employ orthogonal validation methods, such as a combination of laser microdissection,

sample purification, and LC-MS, to confirm the results obtained from MALDI-MS[58]. Despite such compelling in situ characterization of lipid diversity and distribution, the function of specific bioactive lipids within the brain remains largely unexplored. SC-MSI enables in situ exploration and elucidation of cell-type- and region-specific dynamics of the brain lipidome, which in turn may help further decipher the biological functions of these bioactive lipids. In addition, analyzing single cells in their native state can provide more accurate information about cellular biological processes and can offer more relevant data for downstream analysis. We noted that current MS-based single-cell imaging methodologies often involve sample pretreatments, such as buffer washing, or require vacuum conditions during ionization. These sample pretreatments may potentially perturb the native states of the cell, resulting in less accurate and relevant biological data. Therefore, we believe that a promising future direction of single-cell MS imaging is to perform native analysis of single cells, which will pave the way for the development of the next generation of MSI methodologies for single-cell analysis.

## Methods

### Chemicals and materials

For details about the chemicals used in the study, refer to the Supplementary Methods. The commercial human cell lines, including pancreatic cancer cell line PANC-1 (ATCC® CRL-1469) and neuroblastoma cell line (SK-N-SH), were obtained from American Type Culture Collection (ATCC, Manassas, VA, USA). Human primary pancreatic stellate cells (PSCs) were isolated from PDAC tumor specimens resected from patients at the University of Wisconsin Carbone Cancer Center (UWCCC) with informed consent; the procedure of the PSC cell line has been described previously[59,60]. PANC-1 and PSC cells were cultured at 37 °C with 5% $CO_2$ using DMEM:F12 (Hyclone) containing 10% fetal bovine serum (FBS) (Gibco), 1% penicillin−streptomycin solution (Gibco). SK-N-SH cell lines were maintained in Eagle's Minimum Essential Medium (EMEM) (ATCC, Manassas, VA, USA) containing 10% fetal bovine serum (FBS) (Gibco), 1% penicillin−streptomycin solution (Gibco).

Experiments involving animal use were approved by the Animal Care and Use Committee of the University of Wisconsin-Madison. Tissues were harvested from four female wild-type mice (C57/BL6J, age of 40 weeks). For the collection of fresh mouse brain tissues, euthanasia by decapitation was performed, and the whole brain tissue was surgically collected and snap-frozen in liquid nitrogen immediately. The mouse brain tissue samples were stored at −80 °C until use.

### Sample preparation for single-cell MS imaging

Single-cell samples for SC-MS imaging were performed according to the previously reported protocol[61], with a slight modification. Inspired by recent studies[30,35] on single-cell MS imaging, the schematic workflow of sample preparation for SC MALDI-MS imaging was shown in Fig. 1. Cells were trypsinized and dispersed with 0.25% trypsin-EDTA (Gibco) and split 1:5 after reaching 70–80% confluency. The ITO slides (25 × 75 × 1 mm, Delta Technologies, USA) were washed with 70% ethanol and deionized water, respectively, before the single-cell (SC) seeding. The dispersed SCs (~4.5 × 10³ cells/mL) were seeded on the conductive side of the ITO slides, and the cells were cultured for 1–3 days at 37 °C with 5% $CO_2$. The cells would adhere to the ITO slides during the incubation. After incubation, the cells were washed with phosphate buffer saline (PBS) to clean the residual culture media on the cell surface. After washing, the cells were fixed for 15 min with 4% chilled formaldehyde PBS (4 °C). Following the cell fixation step, the SC slides were washed with PBS and 50 mM ammonium acetate aqueous solution three times each solution. It is worth noting that the use of 50 mM ammonium acetate washing can limit the generation of sodium and potassium adducted ions during MSI. The SC slides were dried down in fume hood and black pen marks were drawn on the back of the ITO slide to mark the cell coordinates, which help in tracking the cell position and image registration. After building the cell coordinates, the ITO slide was scanned in bright-field mode under an inverted microscope (TI-S, NIKON, Japan) with a SPOT imaging system (Diagnostic Instruments, Inc., Sterling Heights, MI) to take the optical images of the cells on the ITO slides. Note that at least three technical replicates for the SC monocultures and co-cultures were used. For the MSI of mouse brain tissue, the mouse brain tissue sections were sagittally sectioned at 10 µm thickness using a cryostat (Thermo Fisher Scientific, San Jose, CA, USA) at −20 °C and thaw-mounted onto ITO slides. The tissue section samples were dried in a desiccator and stored at −80 °C until analysis. Note that the mouse brain tissue slides were washed with 50 mM ammonium acetate aqueous solution three times and then dried in a desiccator prior to the MALDI matrix application.

The MALDI matrix was applied on the SC slides and the tissue sections via either a homemade sublimation device or an M5-Sprayer (HTX Technologies, Carrboro, NC, USA). For application of the matrix via sublimation, sublimation of 30 mg of CHCA at 170 °C for 5 min was performed with a pressure of the sublimation chamber at 0.35 Torr. In the case of the DAN matrix, the sublimation temperature was set at 130 °C and the sublimation time at 120 s while keeping other parameters the same. Herein, we used DHB, CHCA, and DAN as MALDI matrices, as they are commonly used for metabolite analysis. It is important to note that the selection and deposition of the matrix could have a significant effect on the quality of results, and this should be taken into consideration for specific types of analyses. In dual-ionization MS imaging, the CHCA matrix residuals were removed by immersing the slide in a 50 mM ammonium acetate aqueous solution for 3 min following the positive-ion mode MS imaging. This washing process was repeated three times, with fresh solutions used each time. After the matrix washing and drying, an optical image of the single-cell slide was recorded again prior to applying a sequential MALDI matrix, and the ion images in positive and negative ion mode were compared to its optical images independently during data analysis. For the use of M5-Sprayer, CHCA (5 mg mL⁻¹) in acetonitrile/water (v/v, 70:30) solution containing 0.1% formic acid was sprayed onto the SC slides at a flow rate of 50 µL min⁻¹ with tracking space of 2 mm for 10 passes. The nozzle temperature was set to 75 °C, the nozzle nitrogen gas pressure was 10 psi, and the moving velocity of the nozzle was 1000 mm min⁻¹. A drying time between each pass was set to 30 s. Slides coated with the DHB matrix were handled according to the same protocol. Initially, the conventional MALDI matrix of CHCA and DHB was carefully deposited onto SCs via conventional spray coating methodology. The result showed higher signal intensity of lipid species over the mass range of 600–900 Da observed from a single PANC-1 cell with CHCA matrix, whereas the mass spectrum obtained with DHB matrix was dominated by the ion signals derived from matrix peaks (Supplementary Fig. 1). However, co-registration of the SC-MSI results and its microscopic image showed slight chemical delocalization for some lipid species (Supplementary Fig. 2). The chemical diffusion might be due to the "wet-coating" of MALDI matrix using a spray coating methodology, which would significantly diminish spatial resolution for SC-MSI. To address this issue, a "dry-coating" methodology based on sublimation was performed with a custom-made sublimation device (Supplementary Fig. 3). It showed that a smaller size of matrix crystal was obtained via sublimation (Supplementary Fig. 4) and sublimation of the MALDI matrix can help reduce the chemical diffusion of cellular lipids (Supplementary Fig. 5).

### Data acquisition and analysis

MALDI-MS imaging experiments were performed on a TimsTOF flex mass spectrometer (Bruker Scientific, LLC, Bremen, Germany) coupled with a SmartBeam 3D 10 kHz frequency tripled Nd:YAG laser (355 nm). The laser settings used were 10 µm diameter circular spot size, with 100 shots per pixel and a raster step size of 10 µm for cell MS imaging.

The laser power was set to 25–45%. The MS imaging data were collected over a mass range of 400–1000 Da for positive-ion detection mode, and the scan range was set to 200–1000 Da for negative mode. Specifically, in dual-ionization MS imaging of SCs, a laser-shots of 50 (5000 Hz) was used with relative lower laser energy ca. 25% in the first positive MSI run using CHCA as matrix; for the subsequent negative imaging run, the 100 shots (5000 Hz) with laser energy of 40% were conducted, using DAN as the MALDI matrix. It is important to note that, in dual ionization of single cells, it is crucial to carefully control the laser energy at an appropriate level and refrain from using excessive laser energy during the first MSI run, as excessive laser energy could damage the cells on the ITO slide and ultimately compromise the performance in the subsequent MSI run. The QTOF-type instrument is equipped with a dual trapped ion-mobility funnel with the highest mobility resolution ($\Omega$) ca. 200, which can perform accumulation and analysis in parallel. The mass resolving power of the instrument is typically 40000 (fwhm) in the lipid mass range, therefore, high mass accuracy could be obtained in MS imaging. MALDI-MS/MS experiments for target lipid ions were performed with a collision voltage of 30–60 eV with precursor ions isolated at a mass window of 1 Da. Other instrumental parameters include: The ramp time was set at 200–400 ms for the TIMS tunnel. TIMS funnel 1 (accumulation) RF of 400 Vpp, TIMS funnel 2 RF (analysis) of 400 Vpp, 1/K0 start at 0.90 V·s/cm$^2$ and end at 1.60 V·s/cm$^2$, collision RF of 800 Vpp, ion transfer time of 50 µs, prepulse storage time of 20 µs, and multipole RF of 500 Vpp. SC-MSI images were visualized using SCiLS Lab Pro (Bruker Scientific, LLC, Bremen, Germany) without data normalization or denoising. Co-registration of MSI images and the optical image of the cells on the slide were performed by combining the cell microscopic image and the MSI image. Region of Interests (ROIs) for the SCs were picked using the SCiLS Lab Pro or a custom-developed MSI Parser program by the co-registration of the MSI image and the optical images. The average mass spectra of the SC ROIs were exported for further statistical analysis. Lipids detected from the SC samples were identified based on the combination of in situ MALDI-MS/MS of target lipids from the SC using a 10 µm diameter circular size laser spotting, as well as conventional lipidomics analysis of cellular lipid extracts via high-performance liquid chromatography (HPLC)-ESI-MS/MS. The details of lipids extraction and HPLC-ESI-MS/MS analysis refer to Supplementary Information. The cell lipidomics data was analyzed by MSDIAL (Version 4.80) (Supplementary Data 4) and LIPID MAPS database and Human Metabolome Database searching.

### Statistical analysis

Statistical analysis of SC data was based on the Uniform Manifold Approximation and Projection (UMAP)[62] and machine learning-driven classification models. The mass spectrometric fingerprints of PSC and PANC-1 SCs were initially subjected to UMAP using the "umap-learn" Python package. Supervised machine learning classifications were processed using scikit-learn models, including support vector machine (SVM), random forest (RF) classifier, and multilayer perceptron (MLP). In brief, the SC data set was split into a training group (60%) and a testing group (40%) randomly for machine learning model generation and validation. Those models were optimized by each iteration and evaluated by the performance of prediction (Supplementary Table 1). Briefly, we randomly partitioned (6:4) the single-cell data set (consisting of 104 cells, equally distributed between PSC and PANC-1 cells) into a training set (62 cells) and a testing set (42 cells) to ensure the learning process was less biased and that the training data set was representative of the original data set. We trained our models using the training data set and validated them using the testing data set. Note that the SC data set was labeled according to cell type, thereby providing the ground truth for validating the classification outcomes. We constructed a confusion matrix for each model to evaluate its performance on the testing data set. The confusion matrices were constructed, and the accuracy, sensitivity, and specificity were calculated for each model, in which Accuracy = [(TP + TN)/(TP + TN + FP + FN)]; Sensitivity = [TP/(TP + FN)]; and Specificity = [TN/(TN + FP)]; T = True, F = False, P = Positive, N = Negative. The scripts are available on request.

A platform, termed MSI Parser, for automated data analysis, was developed to contribute high-throughput capabilities to the presented pipeline (Supplementary Fig. 30). For data analysis, 8 $m/z$ values corresponding to each background (no cells detected) and foreground (cells detected) were assigned and summed. For each pixel, the background sum intensity was subtracted from the coordinating foreground sum intensity. Pixels with a sum intensity greater than or equal to six times the average background pixel intensity were assigned as cells, denoted with an intensity value of 1, and all other pixels were assigned as background, with an intensity value of 0. Following this, a median filter was applied to eliminate non-cellular artifacts. Using the pyimzML python-based package, cells were defined, and their coordinates were extracted. Our current search criteria included a resolution of 7000 and a peak intensity threshold of 2000 on peak picking. Cells were eliminated automatically based on defined area thresholds. Algorithm accuracy was confirmed through a comparison of selected cell regions of interest (ROI) with optical images. Following this validation, the spectrum for each pixel of a given cell was extracted for an $m/z$ of interest and averaged across the area of the cell. This automated package greatly improved the throughput of the overall workflow. Furthermore, a graphical user interface (GUI) was designed to accompany the program, increasing its accessibility. In Supplementary Figs. 31 and 32, a comparison was presented between the performance of ROI picking via SCiLS Lab manually and the MSI Parser tool. This program and its accompanying documentation are open-source and available free of charge at https://github.com/lingjunli-research/Automatic-MSI-Spectra-Extraction.

### Reporting summary

Further information on research design is available in the Nature Portfolio Reporting Summary linked to this article.

## Data availability

The raw HPLC-MS/MS data and demo MSI data for MSI Parser have been deposited in the Figshare database under https://doi.org/10.6084/m9.figshare.23732421.v1. The MS imaging data included in this study are available from the corresponding author upon request. Source data are provided with this paper.

## Code availability

The MSI Parser tool and all codes are available at https://github.com/lingjunli-research/Automatic-MSI-Spectra-Extraction.

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

## Acknowledgements

Aspects of this work were supported in part by NIH grants R01 AG078794, R01 DK071801, RF1 AG052324, R01 DC004428, R01 DC010777, R01 DC019357, R01 NS094154, P01 CA250972, P50 DE026787, and the United States Department of Agriculture (2018-67001-28266). Some of the mass spectrometers were acquired using NIH shared instrument grants S10 OD028473, S10 RR029531, and S10 OD025084. H.Z. and H.L. would like to thank the funding support for a Postdoctoral Career Development Award provided by the American Society for Mass Spectrometry. L.F. was supported in part by the National Institute of General Medical Sciences of the National Institutes of Health under Award Number T32GM008505 (Chemistry–Biology Interface Training Program). We acknowledge Ethan Yang from the Bruker Imaging Demo Apps Team for help on SCiLS Lab. L.L. would like to acknowledge a Pancreas Cancer Pilot grant from the University of Wisconsin Carbone Cancer Center (233-AAI9632), a Diabetes Research Center (DRC) pilot and feasibility grant from Washington University/ University of Wisconsin-Madison (P30 DK020579), a novel methods pilot grant from the University of Wisconsin Institute for Clinical and Translational Research (UL1 TR002373), as well as a Vilas Distinguished Achievement Professorship and Charles Melbourne Johnson Distinguished Chair Professorship, with funding provided by the Wisconsin Alumni Research Foundation and the University of Wisconsin-Madison School of Pharmacy.

## Author contributions

H.Z. and L.L. designed the research; H.Z., Y.L., X.S. and P.H. performed the experiments; H.Z., Y.L., X.S., P.H., L.F. H.L. N.V.W., X.T. and L.L. analyzed data; A.J.S. and L.P. provided the sample; H.Z. and L.L. wrote the paper. All authors contributed to the discussion of the project and have reviewed and approved the final paper.

## Competing interests

The authors declare no competing interests.
