## [Peer Review File · Nature Communications]

Single-cell lipidomics enabled by dual-polarity ionization and ion mobility-mass spectrometry imagingREVIEWER COMMENTS

Reviewer #1 (Remarks to the Author):

General comments:

The manuscript entitled "Single-cell lipidomics enabled by high-throughput mass spectrometry imaging at subcellular resolution" employed trapped ion mobility mass spectrometry coupled with high spatial resolution MALDI-MSI to achieve high-throughput lipidomics analysis of single cells. The authors also developed the dual-polarity ionization MSI method to improve single cell lipidome coverage. Finally, the authors applied the method to several applications, such as cell co-culture samples and mouse brain tissues. MALDI-TIMS-MS is a powerful technique for single-cell analysis. However, the use of high spatial resolution MALDI-MSI for single cell analysis has previously been described by many publications (e.g., <https://doi.org/10.1126/science.abh1623>, <https://doi.org/10.1038/s42255-022-00615-8>, <https://doi.org/10.1016/j.trac.2022.116902>). Additionally, this manuscript did not highlight the advantages of ion mobility separation. The evaluation and demonstration of the advantages of dual-polarity ionization method was insufficient. Overall, the work is of potential interest to the field, but the following major concerns should be addressed before further consideration.

1. The authors should cite and discuss previously publications that used high spatial resolution MALDI-MSI for single cell analysis in introduction part. Further, the novelty and importance of the work should be highlighted.

2. Page 9: I agreed that dual-polarity ionization method can improve the lipidome coverage. However, the procedures of laser ablation and matrix wash could potentially result in the disruption of lipid species and chemical diffusion in the second MALDI analysis. The authors should add comprehensive comparisons between dual-polarity ionization method and single-polarity ionization. Although MALDI-MS technology is efficient to profile large numbers of cells in a single-cell resolution, it also suffers from the poor quantification reproducibility and accuracy. The work did not provide any results related to reproducibility and accuracy. The authors should provide more data to clarify the potential issues.

3. There are many isobaric overlaps among different lipid species, such as [M+2] isotopologue of PC(32:2) and PC(32:1) showed in Figure 2. This would cause inaccurate identification and quantification results. TIMS separation has the potential to improve the differentiations of isomeric and isobaric lipids. However, the authors did not provide detailed effect of TIMS separation in this work. It is better to demonstrate the separation capability of TIMS separation for the common isobars and isomers in the work. Figure 5a demonstrated the TIMS separation of PE(34:0) and PC(O-32:0), which can be distinguished by exact mass as the mass error is more than 50 ppm. It was not a good example for TIMS separation.

4. Page 7 line 131: The authors mentioned that "the identification of lipid species were assigned based on tandem MS by in situ SC-MALDI spotting (Figure S10 and Table S1, S2)". I wonder whether TIMS for multi-dimensional selection of targeted ions was used in the work. TIMS based multi-dimensional selection should be useful for isobars and isomers differentiations.

5. Page 16 Figure 6: Cells were treated with 10 μ M MF-438, which significantly affected the cell proliferation. So, how did authors consider to rule out the effects of cell numbers and viabilities on quantification results of MUFAs. The conclusion was not supported by the data with current experimental design.

6. In Figure 2I, although higher abundance of ceramides was present in the perinuclear region was found with subcellular resolution MALDI-MSI, the authors did not provide any other new information regarding to the application of the high MALDI-MSI resolution. As the authors highlighted the

subcellular resolution in the title, it is better to demonstrate the motivation of subcellular resolution applied in single cell analysis.

7. Some typos: Page 18 line 287: It should be Figure 7d instead of Figure 6d; Table S2: PG 18:0_18:2 and PG 18:1_18:1 appeared twice.

Reviewer #2 (Remarks to the Author):

Zhang et al. present an interesting and well-prepared manuscript on single cell analysis using mass spectrometry imaging. The introduction provides a sound and precise overview of the field and the methods are, for the most part, described in necessary detail. The results are presented in well-composed figures and described appropriately. My main concern with this manuscript is the novelty of the presented results, because most of it has been presented elsewhere already in very similar fashion (see detailed comments below). In addition I find the title a bit misleading. The authors do not present data with clear sub-cellular resolution. Also, while throughput on the MS side may be high, the presented approach to data analysis contains a number of laborious manual steps (e.g. defining ROIs) that slow down the overall process considerably. While I enjoyed reading the manuscript and think that it will be of general interest to the MS imaging community, I would not recommend it for publication in Nature Methods because of the lack in novelty.

Detailed comments:

Results page 5: The preparation of cells for single cell analysis directly on ITO slides has been described in detail by Bien et al. (<https://doi.org/10.1021/acs.analchem.0c04905>). This paper includes detailed comparison of spray coating and sublimation for matrix application as well as washing and fixation steps. The protocol presented in the manuscript is very similar to the existing work that has so far not been referenced.

Figure 1.: The presented workflow is very similar to a figure presented by a different publication by Bien et al. (<https://doi.org/10.1073/pnas.2114365119>). While the work has been cited in this manuscript but in a different context.

Page 9 line 153: From my experience, CHCA is almost insoluble in water. I was wondering how long and how many repetitions of washes with an aqueous solution are needed to remove the CHCA. While novel for single cells, measurements in both polarities with different matrices have been presented for the analysis of tissue. This work should be cited in this context.

Page 10 lines 170 ff: There are a number of publications that use MALDI-MSI to decipher cell-to-cell heterogeneity. These published approaches are very comparable to the work presented here. (e.g. DOI: 10.1126/science.abh1623; <https://doi.org/10.1073/pnas.2114365119>; <https://doi.org/10.1038/s41592-021-01198-0>). All three approaches also use co-registration of microscopy images to identify cellular bodies and ROIs for MS analysis. In this context, the latter two omit laborious manual segmentation of the cells by using segmentation algorithms. Contrary to the claim in lines 172 to 174 most of these also include the analysis of heterogeneity within a single cell line. Again very similar to the presented manuscript, two of these publications also include co-cultured cell lines to demonstrate the use of single cell lipid mass spectra to differentiate cell types using statistical analysis or machine learning. At least one of them also includes the chemical stimulation of a cell line to induce changes in the lipidome.

Page 11 line 177; page 17 lines 263 ff and other places in the manuscript: The authors describe abundances of lipids throughout the manuscript and compare them between cells and tissue regions. I assume that these abundances are directly derived from the measured signal intensities. This direct connection is not permissible, especially for the presented brain data. It has been shown, that signal intensities measured for different lipid species from different regions of the cerebellum do not reflect

the differences in abundance (<https://doi.org/10.1007/s00216-020-02818-y>).

Figure 4: Comparing sub-figures a with b-e, it seems as if the cell bodies identified in the optical images look more narrow than those measured in MSI. In Sub-figure f, the MSI result seems to match the optical image much better. Do you have an explanation for this? How do you differentiate between the two cell types denoted by the red and blue arrows? Is this based on morphology alone?

Figure 5: While the ion mobility separation certainly helps in separation the two lipid ion species, for this example, similar images could be derived from MS separation alone.

Page 17 line 263 ff: I disagree with the authors on their claim that they image on single cell level (also see page 20 line 339). Images of mouse brain at 10 μm pixel size have been presented in a number of publications with the same quality and can be considered standard. While the authors show evidence of cellular layers, they are not able to resolve single cells or correlate their findings with distinct features in the optical images. MS-imaging of cellular layers on the other hand is not new, but similar work has been shown in the literature, for example in the context of eyes. (<https://doi.org/10.1007/s13361-014-0883-2>).

Methods:

Page 25 line 424: A total ion count normalization does not seem meaningful in the context of single cell analysis. The area in between cells is more or less empty and will produce much less TIC in the measured m/z -range. Therefore, normalization will lead to the formation of artifacts. Small signal intensities produced from small amounts of lipids leaked from cells into the surrounding, for example, will be amplified by TIC normalization in areas of low overall ion signal and distort the cellular morphology.

Page 25 line 425: Please describe the selection process of ROIs in more detail. What was used as cellular boundary? How much bias do you expect from manual labeling? How many cells were labeled and were all cells from a specific area included?

Page 25 line 434 ff: The description of the statistical analysis needs to be more detailed. For the machine learning part: What did you consider ground truth and how was the ground truth derived? Did you apply the machine learning to the same data it was trained on, or was it tested on an independent data set like a technical replicate? How many cells did you use for training and were all cells from a specific area included in the data set?

Reviewer #3 (Remarks to the Author):

The authors describe a method using MALDI mass spectrometry imaging with a highly focused laser to detect lipids in single cells, cultured and in tissue. To avoid diffusion of lipids, they use matrix sublimation and to limit the amount of sodium and potassium they wash their samples with ammonium acetate after PBS, even though this is not specifically stated. The most interesting thing is that the authors describe the use of dual polarities from one individual cell to further broaden the types of lipids that are detected. Especially, since this can open up for simultaneous studies of many different pathways. The authors also use a trapped ion mobility separation that has the potential of revealing in depth chemical information, but the one example shown from its implementation suggests that separation was already achieved by the difference in m/z between the two lipids. Overall, it is a nice manuscript but it lacks some validation and differentiation to previously published and seemingly similar studies.

Major comment

Nowhere in the manuscript are any spectra shown from standards. Although it is commonly known

that phospholipids are highly abundant in cells and that these are likely the species detected, it would further validate the study to show that lipid standards subjected to the same treatment, several washing steps, fixation, and more washing, are still intact and that the relative abundance is accurate. Please show validation using standard mixtures.

The dual polarity mode would also benefit from validation with standards to ensure that the steps in between and the large amount of energy put into the small space don't alter the speciation, or increase fragmentation. To ensure that there are no difference in the reanalysis of the cells, the authors should include data showing that reanalysis in positive ion mode provides the same results as the first analysis. Please include these experiments.

There is a difference between mass accuracy and mass resolution. The TIMS-TOF instrument has a modest resolution, stated at 20 000 at m/z 600, which does not necessarily warrant annotation by mass alone. There are a few MSMS spectra in the SI for the manuscript, but there are annotations in the SI tables that are 10 ppm away. This suggests that these may not be the actual species. Please include data confirming these annotations.

Additionally, it seems like the authors are assuming that they have only protonated species present in the mass spectra. However, there are some masses that overlap between protonated and sodiated species, where the sodiated annotation would make more sense. I advise that the authors combine the use of standards and endogenous species and combine this with their trapped ion mobility feature and MSMS to ensure that the annotations are indeed accurate and that there are no sodiated or potassiumated peaks in the mass spectra. For example, the ion at 782.5685 is more likely a sodiated PC 34:1.

How do the authors know that the differences in PL species are because of the cells and not the methodology? A cell can grow more flat or more compact on the surface. Depending on the penetration depth of the laser, it is safe to assume that lipids sampled from a more compact cell would have less influence of plasma membrane lipids than lipids sampled from a cell that is flat on the surface. Please compare cells of the same size and structure to show that the differences are indeed from biological differences and not methodological.

Before analysis the cells are exposed to so many different treatments. If the authors want to find biologically relevant data, would it not be better to analyze cells in their native state? Please elaborate.

Apart from the dual polarity analysis and different cells, it is not clear how this methodology differ from imaging of tissue with high spatial resolution (small laser spot size) MALDI and single cell studies with MALDI (including the SpaceM mentioned in the manuscript) that has been published previously (For example DOI 10.1007/s00418-013-1097-6, DOI: 10.1007/s13361-014-0883-2, DOI:10.1038/s41592-021-01198-0). Please specify.

Minor comments

In Figure 5 the authors detail how the implementation of ion-mobility improved the separation of isobaric compounds 720.5497 and 720.5866 m/z and resulted in dramatically different distributions of these two lipids between PANC1 and PSC cells. However, as can be seen in Figure 5 a) the two ions can be sufficiently separated by MS as well. Have the authors tried to depict the spatial distribution of these two lipid species based on solely the m/z data? It would be interesting to see a comparison between the UMAP plots of PANC1 and PSC cell subpopulations with and without using CCS data.

Do the authors have any data on CCS values of structural isomers or were they indistinguishable by ion-mobility MS?

I generally miss the information on spatial resolution for the different single-cell images. Some images seem to have higher resolution than others, for example, images in Figures 2 and 3 seem to have higher spatial resolution than those in Figure 4. It would be nice to include details on spatial resolution either in the text or in the image descriptions, as the authors did in the case of brain tissues. I also didn't find any data about cell sizes which would be useful to include as well to get a deeper insight into the extent of subcellular resolution.

In Supplementary Figure S4 when comparing the MALDI matrix depositing techniques, the authors are using different magnification values in each case, which makes them hardly comparable. Please adjust to the same magnification.

Please include a motivation to the choice of matrix and why only two matrices were tested.

In the Figures the authors state that the displayed ion images is of lipids annotated within 5 ppm, but in the SI tables the error is much larger. How many of the 180 lipids that the authors claim to have annotated are actually within the 5 ppm? Please specify.

In Figure 4 it is unclear what cells that are of which kind. Please include a specification on how the authors know what cell is PANC1 and PSC.

In Figure 6 it seems like treatment above 1 μM has significant changes, however, the authors state above 10 μM . Please explain your reasoning for choosing this higher number.

REVIEWER COMMENTS

Reviewer #1 (Remarks to the Author):

General comments:

The manuscript entitled “Single-cell lipidomics enabled by high-throughput mass spectrometry imaging at subcellular resolution” employed trapped ion mobility mass spectrometry coupled with high spatial resolution MALDI-MSI to achieve high-throughput lipidomics analysis of single cells. The authors also developed the dual-polarity ionization MSI method to improve single cell lipidome coverage. Finally, the authors applied the method to several applications, such as cell co-culture samples and mouse brain tissues. MALDI-TIMS-MS is a powerful technique for single-cell analysis. However, the use of high spatial resolution MALDI-MSI for single cell analysis has previously been described by many publications (e.g., <https://doi.org/10.1126/science.abh1623>, <https://doi.org/10.1038/s42255-022-00615-8>, <https://doi.org/10.1016/j.trac.2022.116902>). Additionally, this manuscript did not highlight the advantages of ion mobility separation. The evaluation and demonstration of the advantages of dual-polarity ionization method was insufficient. Overall, the work is of potential interest to the field, but the following major concerns should be addressed before further consideration.

Response: We appreciate the reviewer’s critical, insightful, and constructive comments on our manuscript.

We are much obliged to the reviewer’s positive comment toward the dual-polarity ionization method for single-cell MSI that we developed and demonstrated in this work. With the reviewer’s comments and major concerns, we have performed additional necessary experiments, and added substantial supporting data into the revised manuscript to strengthen the novelty of this work: 1) we highlight the advantages of ion mobility separation in the MS imaging of single-cell lipidome; 2) we built a user-friendly graphical user interface (GUI) for single-cell data analysis to achieve more automated single-cell region picking and statistical analysis; 3) as a proof-of-concept study, we introduced in-situ single-cell epoxidation chemical derivatization for in-depth analysis of lipid C=C isomers; 4) we substantially revised our manuscript with more detailed discussions based on the additional experiments. Overall, we believe the quality of the manuscript has been substantially improved with the generous help of this reviewer.

1.The authors should cite and discuss previously publications that used high spatial resolution MALDI-MSI for single cell analysis in introduction part. Further, the novelty and importance of the work should be highlighted.

Response: We thank the reviewer for pointing out this issue and we sincerely appreciate these suggestions. Previous publications that used high spatial resolution MALDI-MSI for single-cell analysis have been cited and discussed in the manuscript. We also highlighted the novelty and importance of this work in the introduction, as below:

“Given that the spatial resolution of MALDI-MSI has advanced to the submicron level as a result of instrumentation advances including atmospheric pressure MALDI³⁶, transmission-mode MALDI, and laser

post-ionization (t-MALDI-2)³³, subcellular resolution MSI is now considered methodologically feasible. For instance, with a novel optics system, atmospheric pressure MALDI³⁶ was able to achieve a lateral resolution of 1.4 μm , which could be further reduced to 600 nm for brain tissue via transmission-mode MALDI and laser post-ionization (t-MALDI-2)³³. The lack of a molecular separation procedure following metabolite desorption/ionization presents an additional challenge to current SC MALDI-MSI approaches, due to a considerable ion suppression effect. This challenge is particularly relevant to SC lipidomics, as lipids exhibit high structural complexity resulting from the presence of numerous isobaric/isomeric species^{38, 39}. There is still a strong demand for higher molecular specificity in SC MALDI-MS imaging.”

We also highlighted the novelty and importance of this work in the introduction, as below:

“In this study, we implemented trapped ion mobility separation combined with dual-polarity ionization MSI to enable high throughput *in situ* profiling of SC lipidomes. We first evaluated the performance of this approach using MSI analysis of SC samples from mono- and co-cultured human pancreatic cancer (PANC-1) and activated pancreatic stellate cells (PSC), as well as neuroblastoma cells (SK-N-SH); analyses were performed using a MALDI trapped ion mobility time-of-flight/time-of-flight MS (MALDI-timsTOF-MS) platform. Further, as a proof-of-concept study, we also introduced in-situ SC epoxidation chemical derivatization for in-depth characterization of the unsaturated lipid isomers.”

We also added discussion about the novelty and importance of this work in the Discussion section, as below:

“Indeed, recent advances have shed light on important aspects of SC-MS analysis and have significantly advanced our understanding of the underlying molecular heterogeneity in SC^{4, 9, 13, 14, 35}. While our study shares a similar sample preparation in SC-MS imaging compared with previous publications^{30, 35}, the current study has several unique merits. First, a dual-ionization strategy for SC MS imaging on the same individual cells were developed, which significantly enhanced the lipidome coverage from a single cell. Second, incorporating ion mobility separation in SC-MSI provides a distinct advantage in revealing isobaric and isomeric lipid species. To the best of our knowledge, the application of ion mobility has not been reported in previous SC-MSI studies. Moreover, a novel graphical user interface (GUI) platform was introduced for single-cell MSI data analysis, offering streamlined and automated single-cell MSI data analysis. Furthermore, as a proof-of-concept study, we introduced *in-situ* single-cell epoxidation chemical derivatization for in-depth analysis of lipid C=C isomers. These advances demonstrate the potential of our approach to provide valuable insights into the lipid heterogeneity of single cells.”

2.Page9: I agreed that dual-polarity ionization method can improve the lipidome coverage. However, the procedures of laser ablation and matrix wash could potentially result in the disruption of lipid species and chemical diffusion in the second MALDI analysis. The authors should add comprehensive comparisons between dual-polarity ionization method and single-polarity ionization. Although MALDI-MS technology is efficient to profile large numbers of cells in a single-cell resolution, it also suffers from the poor quantification reproducibility and accuracy. The work did not provide any results related to reproducibility and accuracy. The authors should provide more data to clarify the potential issues.

Response: We appreciate the reviewer for this helpful comment! We provide our detailed responses below.

(1) Following the suggestion, we compared the dual-polarity ionization method and single-polarity ionization for the SC-MSI. Briefly, for the single-polarity ionization experiment, the PSC cells on the ITO slide were MS imaged under negative mode with DAN matrix directly; while for the dual-polarity ionization experiments, the PSC cells were first MS imaged under positive mode using the CHCA matrix, then the CHCA matrix were washed away with 50 mM ammonium acetate aqueous and the same PSC cells were subjected to negative mode MSI using the DAN matrix. The DAN matrix application condition was identical in both scenarios. As depicted in **Figure S16**, the mass spectra exhibit a high degree of consistency between the single-polarity ionized and dual-polarity ionized PSC cells though the signal intensities of some lipid species were slightly reduced in the dual-polarity ionization scenario. It's important to note that during the first run of MSI on single cells, it is crucial to control the laser energy and avoid using excessive energy levels. High laser energy can potentially damage or even burn out the cells on the ITO slide, leading to decreased performance in the second MSI run. Careful control and management of laser energy is necessary to ensure optimal results in subsequent runs.

We understand the reviewer's concern about matrix washing after the first run of MSI could potentially result in the disruption of lipid species and chemical diffusion in the second MALDI MSI analysis. Our additional experiments showed that the lipid species detected after the first-MSI-run and matrix-washing were quite consistent with that obtained from cells without these pretreatments (**Figure S16**); and the MS images obtained from the second MSI suggested that the extent of chemical delocalization caused by ammonium acetate washing is generally considered to be minimal. (**Figure S17** and **Figure 4**). In our experiment, the CHCA matrix was washed away using 50 mM ammonium acetate. It is noted that ammonium acetate aqueous is a widely used washing solution in MALDI-MSI. This is because ammonium acetate is a mild salt that does not react strongly with most substances, and it does not cause significant changes in the chemical properties or structure of the substance being washed. This has been confirmed by an earlier study that the use of ammonium acetate (50 mM) to wash mouse brain tissue sections can result in a noted improvement in signal intensity and lipidome coverage during negative ion mode MALDI-MS analysis, without any indication of delocalization at a lateral resolution of 10 μm (Richard Caprioli et al, *Anal. Chem.* 2012, 84, 3, 1557–1564).

Figure S16. Mass spectra obtained from PSC SC samples: (a) negative ionization mode mass spectra obtained from PSC cells undergo single-polarity ionization, (b) negative ionization mode mass spectra obtained from PSC cell undergo dual-polarity ionization (1st positive MSI using the CHCA matrix, followed by negative MSI with DAN matrix), (c) zoom-in mass spectra of spectra a, (c) zoom-in mass spectra of spectra b. Here we compared the negative mode mass spectra obtained from PSC cells that without prior-positive-MSI and the mass spectra of PSC cells with dual-polarity ionization (positive mode MSI using the CHCA matrix followed by negative mode MSI with DAN matrix).

Figure S17. Sequential MS imaging of the same single cells using the dual-ionization strategy: (a) bright-field image of the PSC cells on the ITO slide prior to MS imaging, (b) bright-field image of the PSC cells after the positive ionization mode MS imaging (the cells were coated with CHCA matrix and the blue dash rectangle indicated the MSI area), (c) microscope optical image of the PSC cells after the CHCA matrix being washed away using 50 mM ammonium acetate, (d)-(g) representative MS images of the PSC cells obtained via the positive mode ionization SC-MSI, (h)-(k) representative MS images of the same PSC cells obtained from the subsequent negative mode ionization SC-MSI. All SC-MSI images were obtained with mass error tolerance of 10 ppm. Scale bar, 400 μm .

Following the reviewer's comment, we added the additional experiment results and discussions into the revised manuscript, as below:

"As shown in **Figure S16**, the mass spectra exhibit a high degree of reproducibility as reflected from the consistency between the single-polarity ionized and dual-polarity ionized PSC cells, though the signal intensities of some lipid species were slightly reduced in the dual-polarity ionization scenario. Furthermore, the MS images obtained from the second MSI suggested that the delocalization of the lipid species was relatively minimal (**Figure S17** and **Figure 4**)."

“It is important to note that, in dual ionization of single cells, it is crucial to carefully control the laser energy at an appropriate level and refrain from using excessive laser energy during the first MSI run, as excessive laser energy could damage the cells on the ITO slide and ultimately compromising the performance in the subsequent MSI run. In our experiment, the CHCA matrix was washed away with 50 mM ammonium acetate after the first MSI run. The extent of lipids delocalization caused by ammonium acetate washing is generally considered to be minimal. This has been confirmed by an earlier study that the use of ammonium acetate to wash mouse brain tissue sections can result in a noted improvement in signal intensity and lipidome coverage during negative ion mode MALDI-MS analysis, without any indication of delocalization at a lateral resolution of 10 μm ⁵².”

(2) With the reviewer’s suggestion, additional experiments using lipid standards were performed to evaluate the reproducibility and accuracy of SC-MALDI-MSI. Briefly, the blank ITO slide was pre-coated with DAN matrix via sublimation, then, the lipid standards of LPC 18:1 and PC (18:1-18:1) at a concentration of 0.1 $\mu\text{g mL}^{-1}$ in 70% ACN were sprayed onto the surface of the matrix. Thus, the lipids standards were relatively homogeneous deposited on the ITO slide. The working parameters of the MS-Sprayer were set as follows: a flow rate of 20 $\mu\text{L min}^{-1}$, tracking space of 2 mm, 8 passes, nozzle temperature of 60 °C, nozzle nitrogen gas pressure of 10 psi, and moving velocity of the nozzle was 1000 mm min^{-1} , drying time between each pass was set to 30 s.

Then the lipid standard slide was subjected to MALDI-MSI analysis under the identical parameters of single-cell MS imaging, in which 12 separate square regions (ca. 200 $\mu\text{m} \times 200 \mu\text{m}$, 400 pixels of each) on the lipid standard slide were selected for MS imaging. A box plot of the signal intensities of LPC 18:1 ($[\text{M}+\text{H}]^+$, m/z 522.3545) and PC (18:1-18:1) ($[\text{M}+\text{H}]^+$, m/z 786.6007) from each spectrum/pixel were shown in **Figure S6**. Among the 12 regions, the relative standard deviations (RSDs) were 9.5%–10.5% for the signal intensities of LPC 18:1 and the RSDs were 11.8%–14.3% for PC (18:1-18:1) (**Figures S6a and b**), while the average intensities of LPC 18:1 and PC (18:1-18:1) among the 12 regions were quite consistent, with RSDs ($n = 12$) of 1.35% and 1.67% for LPC 18:1 and PC (18:1-18:1), respectively. Also, the average mass spectra from the 12 regions showed that the peaks of LPC 18:1 and PC (18:1-18:1) exhibited a good alignment between different regions as no mass shift was found (**Figures S6c and d**). The results indicate that the SC-MALDI-MSI could offer good reproducibility and mass accuracy for single-cell analysis. Indeed, signal intensity variations were found within each region, this might be largely due to the fact that the deposition of lipid standards was not fully homogeneous using a pneumatic sprayer and the lipid solution microdroplets could affect the MALDI matrix crystal upon its landing on the surface.

We added these experimental results in the revised manuscript, as below:

“To further evaluate the reproducibility and accuracy of MALDI-MSI for SC analysis, lipid standards of LPC 18:1 and PC (18:1-18:1) were deposited onto a matrix coated slide via automated sprayer, and 12 square regions (ca. 200 $\mu\text{m} \times 200 \mu\text{m}$, 400 pixels of each) from the slide were selected for MS imaging. Among the 12 regions, relative standard deviations (RSDs) for LPC 18:1 ($[\text{M}+\text{H}]^+$, m/z 522.3545) were 9.5%–10.5% and the RSDs were 11.8%–14.3% for PC (18:1-18:1) ($[\text{M}+\text{H}]^+$, m/z 786.6007), while the average intensities of LPC 18:1 and PC (18:1-18:1) across the 12 regions were quite consistent, with RSDs ($n = 12$) of 1.35% and 1.67% for LPC 18:1 and PC (18:1-18:1), respectively (**Figures S6a and b**). Also, the average mass spectra from the 12 regions showed that the peaks of LPC 18:1 and PC (18:1-18:1) were in a good alignment

(Figures S6c and d). The results indicate that the high resolution MALDI-MSI could offer good reproducibility and mass accuracy for SC analysis. Indeed, signal intensity variations were found within each region, this could be due to not complete homogenous deposition of the lipid standards onto the slide surface by using a pneumatic sprayer and the lipid solution microdroplets could also affect the MALDI matrix crystal upon its landing on the surface of the slide.”

Figure S6. Box plot of signal intensities of lipid standards from 12 regions on the ITO slides: (a) LPC 18:1 ($[M+H]^+$, m/z 522.3545), (b) PC (18:1-18:1) ($[M+H]^+$, m/z 786.6007); (c) overlapped peaks of LPC 18:1 ($[M+H]^+$, m/z 522.3545) from the average mass spectra of each region (n = 12), (d) overlapped peaks of PC (18:1-18:1) ($[M+H]^+$, m/z 786.6007) from the average mass spectrum of each region (n = 12), the green shaded rectangular background indicates a mass region of 10 ppm. Among the 12 regions, relative standard deviations (RSDs) for LPC 18:1 ($[M+H]^+$, m/z 522.3545) were 9.5%–10.5% and the RSDs were 11.8%–14.3% for PC (18:1-18:1) ($[M+H]^+$, m/z 786.6007), while the average intensities of LPC 18:1 and PC (18:1-18:1) across the 12 regions were quite consistent, with RSDs (n = 12) of 1.35% and 1.67% for LPC 18:1 and PC (18:1-18:1), respectively.

3. There are many isobaric overlaps among different lipid species, such as [M+2] isotopologue of PC(32:2) and PC(32:1) showed in Figure 2. This would cause inaccurate identification and quantification results. TIMS separation has the potential to improve the differentiations of isomeric and isobaric lipids. However, the authors did not provide detailed effect of TIMS separation in this work. It is better to demonstrate the separation capability of TIMS separation for the common isobars and isomers in the work. Figure 5a demonstrated the TIMS separation of PE(34:0) and PC(O-32:0), which can be distinguished by exact mass as the mass error is more than 50 ppm. It was not a good example for TIMS separation.

Response: (1) We thank the reviewer for this helpful comment. We agree with the reviewer that there are many isobaric overlaps among the lipid species. We carefully double-checked the data in **Figure 2**, in the mass spectra of PSC cells, the peak at m/z 732.554 (PC (32:1), $[M + H]^+$) is a dominant peak, and the peak of m/z 730.538 (PC (32:2), $[M + H]^+$) was at low abundance. Theoretically, as shown in **Figure S27**, the accurate m/z of [M+2] isotopologue for PC(32:2) ($[M + H]^+$) is 732.54451 Da with a relative abundance of 11.45% corresponding to its monoisotopic peak; while the monoisotopic mass for PC(31:1) ($[M + H]^+$) is 732.55378 Da. There is a 13 ppm mass difference between the [M+2] isotopologue of PC(32:2) (m/z 732.54451) and the PC(31:1) monoisotopic (m/z 732.55378). Thus, the interference caused by the [M+2] isotopologue of PC(32:2) (m/z 732.54451) is small in the current experiments, as the PC (32:1) ($[M + H]^+$, m/z 732.55378) is a dominant peak and the two ions could be distinguished owing to their mass difference (13 ppm).

Following the reviewer's point, we added discussion and **Figure S27** into the revised manuscript as below:

"It is noted that SC-MSI also requires high mass resolution to avoid the isobaric overlaps among different lipid species and careful consideration when construct the MS images of SC (**Figure S27**)."

Figure S27. SC-MSI requires high mass resolution to avoid the isobaric overlaps among different lipid species and careful consideration when construct the MS images of SC, using the isotopic pattern of PC(32:2) (a) and PC(32:1) (b) as an example: theoretically, the accurate m/z of [M+2] isotopologue for PC(32:2) ($[M + H]^+$) is 732.54451 Da with a relative abundance of 11.45% corresponding to its monoisotopic mass; while the monoisotopic mass for PC(32:1) ($[M + H]^+$) is 732.55378 Da. There is a 13 ppm mass difference between the [M+2] isotopologue of PC(32:2) (m/z 732.54451) and the PC(32:1) monoisotopic (m/z 732.55378).

(2) We appreciate the reviewer for raising this point about the Tims data. We performed additional experiments with IM-MS, and we noticed that more lipid species could be revealed from single cells with the assistance of ion mobility separation. As shown in **Figure 3**, while only one peak was observed when using MS alone, the incorporation of ion mobility separation enabled the recognition of four distinct lipid species. The result further demonstrates that ion mobility provides a distinct advantage in revealing isobaric and isomeric lipid species in SC-MSI. Following this helpful suggestion, new data and discussion have been added into the revised manuscript to demonstrate and highlight the advantage of ion mobility separation for lipid isobars and isomers as follows.

Figure 3. SC-MSI of the PSC cells coupled with the ion mobility separation: (a) representative ion-mobility MS incorporated data at a mass window of m/z 762.5 to 762.8 Da, (b) MS images constructed based on m/z value alone, (c-f) MS images of isobaric and isomeric lipids revealed based on an incorporation of the m/z value and collisional cross section (CCS) values information. Scale bar, 400 μ m.

A description of this new data has been added into the revised manuscript accordingly, as below:

“We also noticed that more lipid species could be revealed from SC with the assistance of ion mobility separation, as illustrated in **Figure 3**, while only one peak was observed when using MS alone, the incorporation of ion mobility separation enabled the resolution and recognition of four distinct lipid species. In addition, variations in lipid isobaric/isomeric composition between the PSC and PANC1 cells

were observed with ion mobility separation (**Figures S10, S11**). These results demonstrate that ion mobility provides a distinct advantage in revealing isobaric and isomeric lipid species in SC-MSI.”

Furthermore, we also added additional Tims data into the **Supporting Information**, showing the lipid difference between the PSC and PANC1 cells revealed using SC-MALDI-MSI coupled with ion mobility separation. The data are shown as below:

Figure S10. Average ion mobility heat maps and mass spectra obtained from the PSC cells (a) and PANC1 cells (b) using SC-MALDI-MSI coupled with ion mobility separation under positive ionization mode. The heat map displays a diverse range of ions for each m/z value, showing enhanced peak capacity achieved through the integration of the TIMS separation dimension. The mass spectral ranges include m/z values 700–850 and the heat map incorporates $1/K_0$ values 1.35–1.49.

Figure S11. Representative zoom-in ion mobility heat maps and mass spectra obtained from the PSC cells and PANC1 cells using SC-MALDI-MSI coupled with the ion mobility separation under positive ionization mode. (a) PSC cells at the mass spectral range of m/z 762.4–762.8, (b) PANC1 cells at the mass spectrum

range of m/z range 762.4–762.8, (c) PSC cells at the mass spectral range of m/z at 788.4–788.8, and (d) PANC1 cells at the mass spectral range of m/z 788.4–788.8.

4. Page 7 line 131: The authors mentioned that “the identification of lipid species were assigned based on tandem MS by in situ SC-MALDI spotting (Figure S10 and Table S1, S2)”. I wonder whether TIMS for multi-dimensional selection of targeted ions was used in the work. TIMS based multi-dimensional selection should be useful for isobars and isomers differentiations.

Response: We thank the reviewer for this insightful point. In this study, we did not utilize the TIMS for multi-dimensional selection of targeted ions for tandem MS identification. We fully agree with the reviewer’s suggestion that TIMS based multi-dimensional selection should be useful for isobars and isomers differentiations. As we know, the TIMS employs parallel accumulation–serial fragmentation (PASEF) to accumulate precursors and subsequently release the precursor ions for MS/MS in a particular order depending on the mobility characteristics. This feature provides a unique advantage for the analysis of isobaric and isomeric species. Unfortunately, the PASEF module is not currently available on our MALDI-timsTOF-MS platform and is only available for LC-MS/MS analysis. Another reason for not using TIMS in tandem MS of the target lipids is that the sensitivity for the fragment ions tends to decrease when TIMS is employed.

For further identifying the lipid species in single cells, the cellular lipid extracts were also subjected to HPLC-nanoESI-MS/MS analysis using the PASEF mode on the TimsTOF-MS platform, in addition to Orbitrap mass spectrometry data. The ion mobility information of the lipids has been added to Tables S1 and S2 in this revision. This combined approach allows for a more comprehensive analysis of the cellular lipids, with Orbitrap mass spectrometry platform providing high-resolution mass measurements of lipids, while HPLC-nanoESI-MS/MS in PASEF mode facilitates the characterization of lipid structures.

5. Page 16 Figure 6: Cells were treated with 10 μ M MF-438, which significantly affected the cell proliferation. So, how did authors consider to rule out the effects of cell numbers and viabilities on quantification results of MUFAs. The conclusion was not supported by the data with current experimental design.

Response: We thank the reviewer for raising this point. MUFAs data in Figure 7 (previous Figure 6) are normalized by the number of cells. The relative intensities of the MUFAs were measured from each individual cell ($n = 50$ for both the drug-treated and control groups), so the cell numbers do not affect the quantification results.

We apologize for this ambiguous description in Figure 7, and we revised the y-axis label “relative intensity ratios (R. Ins)” in Figure 7 f-k to “relative intensity ratios per cell (R. Int. per-cell) to clarify this. The updated figure is below:

Figure 7. SC-MSI analysis of PSC cells under MF-438 drug treatment

6. In Figure 2l, although higher abundance of ceramides was present in the perinuclear region was found with subcellular resolution MALDI-MSI, the authors did not provide any other new information regarding to the application of the high MALDI-MSI resolution. As the authors highlighted the subcellular resolution in the title, it is better to demonstrate the motivation of subcellular resolution applied in single cell analysis.

Response: We thank the reviewer for this insightful point! We strongly believe that subcellular resolution single-cell MS imaging will serve as a premier technology in single-cell study, especially in revealing the roles of bioactive molecules within the individual cells. By providing high spatial resolution information, subcellular resolution single-cell MS imaging can offer novel insights into intracellular biological interactions and reveal previously inaccessible information about cellular heterogeneity. We have added more discussion about this point in the revised manuscript. Also, in response to a comment raised by the second reviewer that the original title could be misleading and to reflect the content of our work more accurately, we changed our title from “Single-cell lipidomics enabled by high-throughput mass spectrometry imaging at subcellular resolution” into “Single-cell lipidomics enabled by dual-polarity ionization and ion mobility-mass spectrometry imaging”.

We added the discussion as below:

“It is noted that subcellular resolution single-cell MS imaging has the potential to become a powerful tool for elucidating the roles of bioactive molecules within individual cells, offering new insights into intracellular biological interactions, and revealing previously inaccessible information about cellular

heterogeneity. Figures 2 and S9 suggest that ceramides were present in the perinuclear regions, indicating that near-subcellular resolution MSI is achieved with these single cell analyses. Indeed, highly sensitive and high-spatial-resolution MSI techniques are still in demand for achieving subcellular resolution MSI.”

7. Some typos: Page 18 line 287: It should be Figure 7d instead of Figure 6d; Table S2: PG 18:0_18:2 and PG 18:1_18:1 appeared twice.

Response: Thank you for bringing those typos to our attention. We have addressed them accordingly in the revised manuscript.

Reviewer #2 (Remarks to the Author):

Zhang et al. present an interesting and well-prepared manuscript on single cell analysis using mass spectrometry imaging. The introduction provides a sound and precise overview of the field and the methods are, for the most part, described in necessary detail. The results are presented in well-composed figures and described appropriately. My main concern with this manuscript is the novelty of the presented results, because most of it has been presented elsewhere already in very similar fashion (see detailed comments below). In addition I find the title a bit misleading. The authors do not present data with clear sub-cellular resolution. Also, while throughput on the MS side may be high, the presented approach to data analysis contains a number of laborious manual steps (e.g. defining ROIs) that slow down the overall process considerably. While I enjoyed reading the manuscript and think that it will be of general interest to the MS imaging community, I would not recommend it for publication in Nature Methods because of the lack in novelty.

Response: We appreciate the reviewer’s critical, insightful, and constructive comments on our manuscript. To address the reviewer’s comments and major concerns, we have performed necessary experiments, and incorporated additional data into the revised manuscript to strengthen the novelty of this work. We highlight below the major **novel** and **unique** aspects of our work compared to previous published work:

1) We developed and implemented a **dual-ionization strategy for single-cell MS imaging**, which can improve the lipidome coverage and throughput in single cell analysis, greatly enhanced the information that can be extracted from the same single cells.

2) We incorporated **ion mobility separation as an additional dimension of analysis with distinct advantage in the single-cell MS imaging**. By incorporating ion mobility separation into MS analysis, more effective separation and identification of complex biomolecules can be achieved that are difficult to distinguish using MS alone. Studies have demonstrated the power of ion mobility for enhancing MS analysis of biomolecules, particularly for lipidome (e.g., *Nature Chemistry*, 2014, 6, 281–294; *Nature Communications*, 2019, 10, 985). Furthermore, recent research has highlighted the enormous potential of ion mobility-MS imaging in enabling the visualization of the spatial distribution of biomolecules (e.g., *Anal.*

Chem. 2019, 91, 22, 14552–14560; *Anal. Chem.* 2020, 92, 13, 8697–8703; *Anal. Chem.* 2023, 95, 2, 1176–1183). Nevertheless, the use of ion mobility in single-cell MS imaging has not been demonstrated.

3) The revised manuscript includes **a novel developed graphical user interface (GUI) that provides a user-friendly platform for analyzing single-cell MSI data**. This GUI offers more streamlined and automated single-cell MSI data analysis, allowing single-cell mass spectra picking and statistical analysis. We have included a description of the automated ROI selection process in the revised manuscript.

4) As a proof-of-concept study, we introduced **in-situ single-cell epoxidation chemical derivatization** for localization of carbon-carbon double bond and in-depth analysis of lipid C=C isomers at the single-cell level.

Taken together, following the reviewer's comments, we have extensively revised the manuscript by performing additional experiments, incorporating new data, and more detailed discussions. The quality and novelty of the manuscript has been significantly enhanced, and we are grateful for the generous help and insightful comments of the reviewer in this regard. We hope the updated version is now acceptable for publication in *Nature Communications*. Once again, we would like to express our sincere appreciation for the reviewer's insightful comments and suggestions.

In response to the comment that "sub-cellular resolution" in the title could be misleading, the "sub-cellular resolution" has been removed from the title. To reflect the content of our work more accurately and better highlight the novel aspect and unique contribution of this work, we changed our title from "Single-cell lipidomics enabled by high-throughput mass spectrometry imaging at subcellular resolution" to "**Single-cell lipidomics enabled by dual-polarity ionization and ion mobility-mass spectrometry imaging**". In addition, the "subcellular resolution" description in the content has been adjusted accordingly throughout the manuscript.

Detailed comments:

Results page 5: The preparation of cells for single cell analysis directly on ITO slides has been described in detail by Bien et al. (<https://doi.org/10.1021/acs.analchem.0c04905>). This paper includes detailed comparison of spray coating and sublimation for matrix application as well as washing and fixation steps. The protocol presented in the manuscript is very similar to the existing work that has so far not been referenced.

Response: We thank the reviewer for tracking the pioneering work done by Bien and Jens et al. on MALDI-MS imaging of single-cells. It is true that we are very much inspired by their contribution in the single-cell study, and this related reference has been cited as Reference #66 in the Method section of the revised manuscript, as below:

"Single-cell samples for SC-MS imaging was performed according to the previously reported protocol⁶⁶ with a slight modification."

Figure 1.: The presented workflow is very similar to a figure presented by a different publication by Bien et al. (<https://doi.org/10.1073/pnas.2114365119>). While the work has been cited in this manuscript but in a different context.

Response: We thank the reviewer for bringing to our attention the recent pioneering work done by Bien and Jens et al. on single-cell MS analysis using high-resolution MALDI-2-MSI in combination with MALDI-compatible staining and use of optical microscopy. Their recent work has shed light on important aspects of single-cell MS analysis and has significantly advanced our understanding of the underlying molecular heterogeneity in cell culture samples. We have taken their innovative approach (Reference #35) into consideration in revising our manuscript and have cited their work accordingly.

Although our study shares a similar workflow for single-cell MS imaging, the current study differs in several aspects. Herein we focus on the single-cell lipidomic specifically and we introduced several **novel aspects** including: 1) dual-ionization strategy for single-cell MS imaging; 2) incorporating ion mobility separation in single-cell MSI for in-depth single-cell lipidomic analysis; 3) a new GUI platform for single-cell MSI data analysis; 4) furthermore, as a proof-of-concept study, we discussed and provided preliminary data of *in-situ* single-cell epoxidation chemical derivatization for in-depth analysis of lipid C=C isomers.

We cite this relevant work in the Method section:

“Inspired by recent studies^{30,35} on single-cell MS imaging, the schematic workflow of sample preparation for SC MALDI-MS imaging was shown in Figure 1.”

Page 9 line 153: From my experience, CHCA is almost insoluble in water. I was wondering how long and how many repetitions of washes with an aqueous solution are needed to remove the CHCA. While novel for single cells, measurements in both polarities with different matrices have been presented for the analysis of tissue. This work should be cited in this context.

Response: We thank the reviewer for this comment.

1) In our experiment, after the positive-ion mode MS imaging, the CHCA matrix residuals were removed by immersing the slide in 50 mM ammonium acetate aqueous solution for 3 minutes, and the washing was repeated 3 times with fresh solutions. CHCA has a relatively high solubility in organic solvents and a moderate solubility in water (in methanol at ~50 mg/ml, in acetonitrile at ~35 mg/ml, and in water at ~6 mg/ml.), therefore, the CHCA residuals could be washed away by the ammonium acetate solution.

The details about the CHCA matrix washing procedure have been added to the Experimental section of the revised manuscript, as follows:

“In dual-polarity ionization MS imaging, the CHCA matrix residuals were removed by immersing the slide in a 50 mM ammonium acetate aqueous solution for 3 minutes following the positive-ion mode MS imaging. This washing process was repeated three times, with fresh solutions used each time.”

2) Indeed, dual-polarities MS imaging have been presented for the analysis of tissue samples in previous publications, however, it is the first time that dual-polarity is demonstrated and implemented in single-cell MS imaging in this study. Following the reviewer’s suggestion, the previous relevant studies on dual-polarities MSI of tissue samples have been cited in the revised manuscript, as follows:

“This is consistent with previous studies⁴⁷⁻⁴⁹ which showed that substantial enhanced molecular coverage is achieved via dual-polarity ionization MS imaging on the same tissue section.”

Page 10 lines 170 ff: There are a number of publications that use MALDI-MSI to decipher cell-to-cell heterogeneity. These published approaches are very comparable to the work presented here. (e.g. DOI: 10.1126/science.abh1623; <https://doi.org/10.1073/pnas.2114365119>; <https://doi.org/10.1038/s41592-021-01198-0>). All three approaches also use co-registration of microscopy images to identify cellular bodies and ROIs for MS analysis. In this context, the latter two omit laborious manual segmentation of the cells by using segmentation algorithms. Contrary to the claim in lines 172 to 174 most of these also include the analysis of heterogeneity within a single cell line. Again very similar to the presented manuscript, two of these publications also include co-cultured cell lines to demonstrate the use of single cell lipid mass spectra to differentiate cell types using statistical analysis or machine learning. At least one of them also includes the chemical stimulation of a cell line to induce changes in the lipidome.

Response: We thank the reviewer for this insightful comment.

1) We apologize for the misleading context in lines 172 to 174 and it has been changed to “While enormous efforts have been devoted to uncovering the SC chemical heterogeneity across different cell lines^{40, 41}, recent research has revealed that substantial heterogeneity also exists within the same cell population, demonstrating significant variability from cell to cell^{4, 30, 35}.” The studies mentioned by the reviewer have been cited as well.

2) Indeed, we realized the issue of laborious manual segmentation in the data analysis. To address this issue, the revised manuscript includes a newly developed graphical user interface (GUI) that provides a user-friendly platform for analyzing single-cell MSI data. This GUI offers more streamlined and automated single-cell MSI data analysis, allowing single-cell mass spectra picking and statistical analysis. The schematic workflow of the GUI is shown in Figure S30, as below:

Figure S30. A custom-developed graphical user interface (GUI), termed as MSI Parser, provides a user-friendly platform for analyzing single-cell MSI data: (a) snapshot of the MSI Parser interface, (b) workflow for the single-cell MS imaging data analysis. This program and its accompanying documentation are open-source and available at <https://github.com/lingjunli-research/Automatic-MSI-Spectra-Extraction>.

3) We thank the reviewer for mentioning these pioneering works on the single-cell study. We agree that statistical methods such as Principal Component Analysis and machine learning are powerful and widely used for revealing the molecular heterogeneity of single cells. In this regard, the statistical analysis approach used in our work shares similarities with the methods employed in the aforementioned studies. However, we focus on the single-cell lipidomic specifically in this study and we introduced several novel aspects for single cell MS imaging including: 1) dual-polarity ionization strategy for single-cell MS imaging; 2) incorporating ion mobility separation in single-cell MSI for single-cell lipidome analysis; 3) a newly developed GUI platform for single-cell MSI data analysis; 4) furthermore, as a proof-of-concept study, we

discussed and provided preliminary data of *in-situ* single-cell epoxidation chemical derivatization for in-depth analysis of lipid C=C isomers.

Page 11 line 177; page 17 lines 263 ff and other places in the manuscript: The authors describe abundances of lipids throughout the manuscript and compare them between cells and tissue regions. I assume that these abundances are directly derived from the measured signal intensities. This direct connection is not permissible, especially for the presented brain data. It has been shown, that signal intensities measured for different lipid species from different regions of the cerebellum do not reflect the differences in abundance (<https://doi.org/10.1007/s00216-020-02818-y>).

Response: We thank the reviewer for this important comment. We agree with the reviewer that direct connection between the lipid signal intensities and its concentrations is not permissible. We have revised the description throughout the manuscript. In addition, we discussed this issue in the revised manuscript and the reviewer mentioned reference has been cited as Reference #58 in the revised manuscript, as below:

“It is worth mentioning that a direct connection between the lipid signal intensities and its abundance from various regions may not be strictly feasible, as several factors can influence the lipid signal intensities in MALDI-MS, such as ion suppression effect, local salt concentration⁶⁰. Thus, to ensure more accurate quantification analysis, it is necessary to employ orthogonal validation methods, such as a combination of laser microdissection, sample purification, and LC-MS, to confirm the results obtained from MALDI-MS⁵⁸.”

Also, the previous description about the lipid “abundance” in relevant context has been changed to lipid “signal intensity” throughout the manuscript.

Figure 4: Comparing sub-figures a with b-e, it seems as if the cell bodies identified in the optical images look more narrow than those measured in MSI. In Sub-figure f, the MSI result seems to match the optical image much better. Do you have an explanation for this? How do you differentiate between the two cell types denoted by the red and blue arrows? Is this based on morphology alone?

Response: We thank the reviewer for the questions. Both PANC1 and PSC cells were characterized based on their cell surface markers and protein expression by collaborators before use. PANC1 was obtained from a commercial source (ATCC), while PSC was isolated from a pancreatic cancer patient sample. These two cell types are from different lineage and have different morphology. These cells have different shapes and sizes on the ITO slide that can be easily differentiated by morphology. PANC1 cells are typically round with a diameter of 18–25 μm , resembling typical cancer epithelial cells. On the other hand, PSC cells are elongated with protrusions, having larger length to width ratios (Figure S19). The average size of PSC is $114 \times 43 \mu\text{m}^2$, which is typically 2.8 times larger than that of PANC1.

Based on several parameters, such as cell length, width, areas, perimeters cell roundness and circularity, ImageJ particle analysis and MALDI imaging software can differentiate these two cell types with high confidence. In original Figure 4 (changed to Figure 5 in revision), the cells denoted by blue and red arrows in our figures were identified by software based on the mentioned parameters. For instance, PSC cells were identified as having a cell area of 300-800 μm^2 , length to width ratio greater than 1.5, and circularity

less than 0.4. PANC1 cells were identified as having a cell area less than $300 \mu\text{m}^2$, length to width ratio less than 1.4, and circularity between 0.5 and 1.0.

Figure S19. Both PANC1 and PSC cells were characterized based on their cell surface markers and protein expression by collaborators before use, these two cell types were from different lineage and have different morphologies. (Top) Bright field images of PANC1, PSC and PANC1+PSC in coculture. All scale bars were equal to $50 \mu\text{m}$. (Bottom) NIH ImageJ software was used to measure cell length, width, areas and perimeters and derived length to width ratio and circularity when PANC1 and PSC cultured individually. The length to width ratio was roughly 1.2:1 for PANC1 whereas that ratio is 2.6:1 in PSC. We didn't find these parameters change when these two cells cultured together in coculture system. These cells have different shapes and sizes on the ITO slide that can be easily differentiated by morphology. PANC1 cells are typically round with a diameter of $18\text{--}25 \mu\text{m}$, resembling typical cancer epithelial cells. On the other hand, PSC cells are elongated with protrusions, having larger length to width ratios. The average size of PSC is $114 \times 43 \mu\text{m}^2$, which is typically 2.8 times larger than that of PANC1.

A description of the coculture sample has been added into the revised manuscript as below:

“These two cell types are from different lineage and have different morphology (**Figure S19**).”

For the comment that the cells in the optical image of Figure 4 look narrower than those revealed by the MSI. This may be due to the optical image not being accurately scaled to the MS images. We have addressed this issue in the revised manuscript to ensure accurate comparison between the optical and MS images. Another reason for this slight difference might be due to the microscope was not focused perfectly when the optical image was taken.

Figure 5: While the ion mobility separation certainly helps in separation the two lipid ion species, for this example, similar images could be derived from MS separation alone.

Response: We appreciate the reviewer for presenting this point on the Tims data. We performed the additional experiment, and we noticed that more lipid species could be revealed from SC with the assistance of ion mobility separation. As shown in Figure 3, while only one peak was observed when using MS alone, the incorporation of ion mobility separation enabled the resolution and detection of four

distinct lipid species. The result further demonstrates that ion mobility provides a distinct advantage in revealing isobaric and isomeric lipid species in SC-MSI. Following this helpful suggestion, new data and relevant discussion have been added to the revised manuscript to demonstrate the advantage of ion mobility separation for lipid isobars and isomers as follows.

Figure 3. SC-MSI of the PSC cells coupled with the ion mobility separation: (a) representative ion-mobility MS incorporated data at a mass window of m/z 762.5 to 762.8 Da, (b) MS image constructed based on m/z value only, (c-f) MS image of isobaric and isomeric lipids revealed based on an incorporation of the m/z value and collisional cross section (CCS) information. Scale bar, 400 μ m.

A description of this new data has been added to the revised manuscript accordingly, as below:

“We also noticed that more lipid species could be revealed from SC with the assistance of ion mobility separation, as illustrated in Figure 3, while only one peak was observed when using MS alone, the incorporation of ion mobility separation enabled the resolution and detection of four distinct lipid species. In addition, variations in lipid isobaric/isomeric compositions between the PSC and PANC1 cells were observed with ion mobility separation (Figures S10 and S11). These results demonstrate that ion mobility provides a distinct advantage in revealing isobaric and isomeric lipid species in SC-MSI.”

Furthermore, we also added additional TIMS data into the Supporting Information, showing the lipid difference between the PSC and PANC1 cells revealed by SC-MALDI-MSI coupled with ion mobility separation. The data are shown as below:

Figure S10. Average ion mobility heat maps and mass spectra obtained from the PSC cells (a) and PANC1 cells (b) using the SC-MALDI-MSI strategy coupled with ion mobility separation under positive ionization mode. The heat map displays a diverse range of ions for each m/z value, showing an enhanced peak capacity achieved through the integration of the TIMS separation dimension. The mass spectra ranges include m/z values 700–850 and the heat map incorporates $1/K_0$ values 1.35–1.49.

Figure S11. Representative zoom-in ion mobility heat maps and mass spectra obtained from the PSC cells and PANC1 cells using SC-MALDI-MSI coupled with the ion mobility separation under positive ionization mode. (a) PSC cells at the mass spectral range of m/z 762.4–762.8, (b) PANC1 cells at the mass spectrum

range of m/z range 762.4–762.8, (c) PSC cells at the mass spectral range of m/z at 788.4–788.8, and (d) PANC1 cells at the mass spectral range of m/z 788.4–788.8.

Page 17 line 263 ff: I disagree with the authors on their claim that they image on single cell level (also see page 20 line 339). Images of mouse brain at 10 μm pixel size have been presented in a number of publications with the same quality and can be considered standard. While the authors show evidence of cellular layers, they are not able to resolve single cells or correlate their findings with distinct features in the optical images. MS-imaging of cellular layers on the other hand is not new, but similar work has been shown in the literature, for example in the context of eyes. (<https://doi.org/10.1007/s13361-014-0883-2>).

Response: We thank the reviewer for this comment. We would be happy to tone down our claim. Following the reviewer's comment, we have adjusted our description in the MS imaging of mouse brain tissues. The H&E-stained histological image of brain tissue section showed that the nuclei size of brain cells is usually over 5 μm (Figure S22). Indeed, for the mouse brain tissue section, 10 μm pixel size MALDI imaging has the potential for single-cell imaging, but we agree that the current study has the limitations in building the precise correlation of the MS imaging results with each individual cells from the tissue.

The previous description of "Our SC-MSI profiles of mouse brain showed region-specific, cell-cell lipidome heterogeneity within the cerebellar cortex (Figure 7e)." has been changed to the following description:

"The high spatial resolution MSI profiles of mouse brain showed region-specific, cellular layers lipidome heterogeneity within the cerebellar cortex (**Figure 8e**). We noted that for SC-MSI of complex tissue samples, the establishment of a precise correlation between the MS imaging results and each individual cell present in the tissue remains a significant challenge. As such, this will be a crucial area in our future research."

Methods:

Page 25 line 424: A total ion count normalization does not seem meaningful in the context of single cell analysis. The area in between cells is more or less empty and will produce much less TIC in the measured m/z -range. Therefore, normalization will lead to the formation of artifacts. Small signal intensities produced from small amounts of lipids leaked from cells into the surrounding, for example, will be amplified by TIC normalization in areas of low overall ion signal and distort the cellular morphology.

Response: We thank the reviewer for this important comment. Following the reviewer's suggestion, all the MSI images have been reconstructed without normalization.

Page 25 line 425: Please describe the selection process of ROIs in more detail. What was used as cellular boundary? How much bias do you expect from manual labeling? How many cells were labeled and were all cells from a specific area included?

Response: We thank the reviewer for raising this point, as indeed, manual steps involved in data analysis do sacrifice the throughput of the overall work. Inspired by this comment, we have since developed a

stand-alone software platform capable of automating the ROI selection and definition process, followed by spectral extraction. By applying this automated workflow, any potential bias is minimized. We have included an explanation of the automated ROI selection process in the Experimental section of the revised manuscript, as follows:

“A platform for automated data analysis was developed to contribute high-throughput capabilities to the presented pipeline. For data analysis, 8 m/z values corresponding to each background (no cells detected) and foreground (cells detected) were assigned and summed. For each pixel, the background sum intensity was subtracted from the coordinating foreground sum intensity. Pixels with a sum intensity greater than or equal to six-times the average background pixel intensity were assigned as cells, denoted with an intensity value of 1, and all other pixels were assigned as background, with an intensity value of 0. Following this, a median filter was applied to eliminate non-cellular artifacts. Using the pyimzML python-based package, cells were defined, and their coordinates were extracted. Cells were eliminated automatically based on defined area thresholds. Algorithm accuracy was confirmed through comparison of selected cell regions of interest (ROI) with optical images. Following this validation, the spectra for each pixel of a given cell was extracted for an m/z of interest and averaged across the area of the cell. This automated package greatly improved the throughput of the overall workflow. Furthermore, a graphical user interface (GUI) was designed to accompany the program, increasing its accessibility. This program and its accompanying documentation are open-source and available free-of-charge at <https://github.com/lingjunli-research/Automatic-MSI-Spectra-Extraction>.”

Page 25 line 434 ff: The description of the statistical analysis needs to be more detailed. For the machine learning part: What did you consider ground truth and how was the ground truth derived? Did you apply the machine learning to the same data it was trained on, or was it tested on an independent data set like a technical replicate? How many cells did you use for training and were all cells from a specific area included in the data set?

Response: We thank the reviewer for presenting this point. To investigate the heterogeneity of SC in PSC and PANC1 cells, we employed supervised machine learning (ML) classifications on SC data obtained from both cell types. Our study utilized three ML models, namely support vector machine (SVM), random forest classifier (RF), and multilayer perceptron (MLP), and a detailed description of the ML models has been provided in the Statistical Analysis section of the revised manuscript.

The SC data set was labeled according to cell type, thereby providing the ground truth for validating the classification outcomes. It was tested on an independent data set (not the same data it was trained on). Briefly, we randomly partitioned (6:4) the single-cell data set (consisting of 104 cells, equally distributed between PSC and PANC1 cells) into a training set (62 cells) and a testing set (42 cells) to ensure the learning process was less biased and that the training data set was representative of the original data set. We trained our models using the training data set and validated them using the testing data set. We constructed a confusion matrix for each model to evaluate its performance on the testing data set. Our results demonstrated high accuracy for all models, and an example confusion matrix for SVM is presented in Figure 6c. Details on the machine learning models for the single-cell classification have been added as Table S4 in the revised manuscript.

Supplementary Table S4. Details on machine learning models trained and validated for supervised classification on SC data.

Model	Optimized hyperparameters	Performance									
Support vector machine	Regularization parameter: 1.0 Kernel function: rbf	Accuracy: 100%     True label \ Predicted label PANC1 PSC     PANC1 18 0   PSC 0 24   	True label \ Predicted label	PANC1	PSC	PANC1	18	0	PSC	0	24
True label \ Predicted label	PANC1	PSC									
PANC1	18	0									
PSC	0	24									
Random forest classifier	n_estimators: 100 Criterion: gini Maximum depth: none	Accuracy: 100%     True label \ Predicted label PANC1 PSC     PANC1 18 0   PSC 0 24   	True label \ Predicted label	PANC1	PSC	PANC1	18	0	PSC	0	24
True label \ Predicted label	PANC1	PSC									
PANC1	18	0									
PSC	0	24									

Reviewer #3 (Remarks to the Author):

The authors describe a method using MALDI mass spectrometry imaging with a highly focused laser to detect lipids in single cells, cultured and in tissue. To avoid diffusion of lipids, they use matrix sublimation and to limit the amount of sodium and potassium they wash their samples with ammonium acetate after PBS, even though this is not specifically stated. The most interesting thing is that the authors describe the use of dual polarities from one individual cell to further broaden the types of lipids that are detected. Especially, since this can open up for simultaneous studies of many different pathways. The authors also use a trapped ion mobility separation that has the potential of revealing in depth chemical information, but the one example shown from its implementation suggests that separation was already achieved by the difference in m/z between the two lipids. Overall, it is a nice manuscript but it lacks some validation and differentiation to previously published and seemingly similar studies.

Response: We greatly appreciate the overall positive evaluation of our work! We thank this reviewer for their excellent summary of our study and positive comments on our techniques and the unique features of our approach. As detailed below with point-by-point response, we extensively revised the manuscript by incorporating additional validation experiments and more detailed discussions. The quality and novelty of the revised manuscript has been significantly enhanced, and we are grateful for the generous help of the reviewer in this regard.

To differentiate from previously published studies, we highlight novel and unique aspects of our work as follows:

1) We highlight the **dual-polarity ionization strategy for single-cell MS imaging**, which can improve the lipidome coverage in single cell analysis.

2) We incorporated **ion mobility separation as an additional dimension of analysis with distinct advantage in the single-cell MS imaging**. By incorporating ion mobility separation into MS analysis, more effective separation and identification of complex biomolecules can be achieved that are difficult to distinguish using MS alone. Studies have demonstrated the power of ion mobility for enhancing MS analysis of biomolecules, particularly for lipidome (e.g., *Nature Chemistry*, 2014, 6, 281–294; *Nature Communications*, 2019, 10, 985). Furthermore, recent research has highlighted the enormous potential of ion mobility-MS imaging in enabling the visualization of the spatial distribution of biomolecules (e.g., *Anal. Chem.* 2019, 91, 22, 14552–14560; *Anal. Chem.* 2020, 92, 13, 8697–8703; *Anal. Chem.* 2023, 95, 2, 1176–1183). Nevertheless, the use of ion mobility in single-cell MS imaging has not been demonstrated.

3) The revised manuscript includes **a novel developed graphical user interface (GUI) that provides a user-friendly platform for analyzing single-cell MSI data**. This GUI offers more streamlined and automated single-cell MSI data analysis, allowing single-cell mass spectra picking and statistical analysis. We have included a description of the automated ROI selection process in the revised manuscript.

4) As a proof-of-concept study, we introduced **in-situ single-cell epoxidation chemical derivatization** for localization of carbon-carbon double bond and in-depth analysis of lipid C=C isomers at the single-cell level.

Taken together, following the reviewer's comments, we have extensively revised the manuscript by performing additional experiments, incorporating new data, and more detailed discussions. The quality and novelty of the manuscript has been significantly enhanced, and we are grateful for the generous help and insightful comments of the reviewer in this regard. We hope the updated version is now acceptable for publication in *Nature Communications*. Once again, we would like to express our sincere appreciation for the reviewer's insightful comments and suggestions.

Thank you for recognizing the merits of our method--"To avoid diffusion of lipids, they use matrix sublimation and to limit the amount of sodium and potassium they wash their samples with ammonium acetate after PBS, even though this is not specifically stated.". We have stated these specific advantages in the revised manuscript, as below:

"It is worth noting that during SC sample preparation, the use of 50 mM ammonium acetate washing can limit the generation of sodium and potassium adducted ions during MSI. Additionally, sublimation of the MALDI matrix can help reduce the chemical diffusion of cellular lipids."

Following the reviewer's suggestion, we also added discussion about the novelty and importance of this work in the Discussion section, as below:

"Indeed, recent advances have shed light on important aspects of SC-MS analysis and have significantly advanced our understanding of the underlying molecular heterogeneity in SC^{4, 9, 13, 14, 35}. While our study shares a similar sample preparation in SC-MS imaging compared with previous publications^{30, 35}, the current study has several unique merits. First, a dual-ionization strategy for SC MS imaging on the same individual cells were developed, which significantly enhanced the lipidome coverage from a single cell. Second, incorporating ion mobility separation in SC-MSI provides a distinct advantage in revealing isobaric and isomeric lipid species. To the best of our knowledge, the application of ion mobility has not been reported in previous SC-MSI studies. Moreover, a novel graphical user interface (GUI) platform was

introduced for single-cell MSI data analysis, offering streamlined and automated single-cell MSI data analysis. Furthermore, as a proof-of-concept study, we introduced *in-situ* single-cell epoxidation chemical derivatization for in-depth analysis of lipid C=C isomers. These advances demonstrate the potential of our approach to provide valuable insights into the lipid heterogeneity of single cells.”

Major comment

Nowhere in the manuscript are any spectra shown from standards. Although it is commonly known that phospholipids are highly abundant in cells and that these are likely the species detected, it would further validate the study to show that lipid standards subjected to the same treatment, several washing steps, fixation, and more washing, are still intact and that the relative abundance is accurate. Please show validation using standard mixtures.

Response: We thank the reviewer for this insightful point and excellent suggestions on performing additional validation experiments using lipid standard mixtures. Following the reviewer’s suggestion, a lipid standard mixture and the cellular lipid extracts from the ITO slide were prepared for the validation experiments. The lipid standard mixture includes FA 16:1, FA 18:1, LPC 18:1, PC 18:1-18:1, PC 18:0-18:1, PS 18:0-18:1, PE 18:0-18:1 at 1 ug/mL of each.

Cell sample washing and fixing are commonly applied procedures in single-cell MS imaging (e.g., Anal. Chem. 93, 4513–4520 (2021); Nature Methods 18, 799–805 (2021)). Similar as the previous studies, the cells cultured on the ITO slides went through washing or fixing steps including phosphate buffer saline (1x PBS) washes (clean the residual culture media), fixing for 15 min with chilled 4% formaldehyde PBS, and 50 mM ammonium acetate aqueous washes.

The cultured cells were first washed using PBS to clean the residual media. Using PBS to wash cells is a standard practice in cell culture and is generally considered to have minimal impact on the cell lipidome. As we know that, PBS is an isotonic solution that does not cause osmotic shock to the cells and is commonly used for a variety of cell culture applications, such as washing cells before dissociation, transporting cells or tissue samples, and diluting cells for counting.

We used the lipid standards mixture solution as a simulated sample, the sample was treated with 4% formaldehyde PBS and 50 mM ammonium acetate, then the lipids were extracted via Folch's extraction procedure, and subjected to nanoESI MS analysis using the Orbitrap Fusion Lumos mass spectrometer. As a control experiment, the sample was treated with deionized (DI) water instead of ammonium acetate and formaldehyde PBS. The result of the lipid standard mixture experiments is shown in Figure S25, showing that the lipids signal is quite consistent between the washing/fixing treated and the control sample. This result indicates that the washing and fixing pretreatments applied prior to SC-MSI have minimal impact on the tested lipid standard mixture sample.

Figure S25. NanoESI-MS analysis lipid standards mixture samples: (a) and (c) are mass spectra of control sample obtained under positive and negative ionization mode, respectively; (b) and (d) are mass spectra of the washing treated sample obtained from positive and negative ionization mode, respectively. The LPC 18:1, PC 18:1-18:1, PC 18:0-18:1 lipid species were detected as protonated ions and the FA 16:1, FA 18:1, PS 18:0-18:1, PE 18:0-18:1 were detected as deprotonated ions in the spectra. The results indicated that the washing and fixing procedure used in the single-cell MSI sample pretreatments have minimal impact on the lipid detection and cell lipidome.

Additionally, we extracted cellular lipids from the ITO coated slides and analyzed the cellular lipid extract samples with the nanoESI-MS under identical setting. Briefly, two ITO slides were seeded with high-density of PANC1 cells (ca. 2×10^4 cells), the ITO slides were harvested after overnight cell adherence. One cell slide sample was treated with the same protocol as used in SC-MSI, which included PBS washes, fixation for 15 min with chilled 4% formaldehyde in PBS, and 50 mM ammonium acetate aqueous washes. Another cell

slide sample was washed with PBS and ammonium acetate, but without formaldehyde fixation. Then, the cell slides were dried under vacuum. After drying the cell slides under vacuum, lipid extraction was performed using 0.5 mL 50% ACN/MeOH solution. The cellular lipid extracts were diluted 100 times with MeOH and subjected to nanoESI-MS analysis in both positive and negative mode in the Orbitrap MS platform. The result is shown in Figure S26, showing that the lipid signals from the sample treated with 15 min chilled 4% formaldehyde fixing and the control sample are highly consistent in either positive or negative ionization mode. This result also indicates that the washing and fixing pretreatments applied prior to SC-MSI have minimal impact on the lipidome of the cells on the slide.

Indeed, it is worth mentioning that caution is required when fixing the single-cell sample with the formaldehyde PBS solution. The fixing time, temperature, and concentration should be set at a low level to avoid excessive fixing, as the excessive fixing can affect the determination of amine-containing lipids such as phosphatidylethanolamine (PE) and phosphatidylserine (PS).

Figure S26. NanoESI-MS analysis of cellular lipid extracts: (a) positive ion mode mass spectrum of cellular lipid extract from PANC1 cell slides without fixation, (b) positive ion mode mass spectrum of cellular lipid extract from PANC1 cell slides with 15 min fixation using chilled 4% formaldehyde, (c) negative ion mode mass spectrum of cellular lipid extract from PANC1 cell slides without fixation, and (d) negative ion mode mass spectrum of cellular lipid extract from PANC1 cell slides with 15 min fixation using chilled 4% formaldehyde.

In summary, following the reviewer's suggestion we performed the validation experiments, and the data (Figures S25 and Figure S26) have been added to the revised **Supplementary Information**, and additional discussions have been added to the Discussion section, as below:

“Our results indicate that the washing and fixing pretreatments applied prior to SC-MSI have minimal impact on the lipidome of the cells on the slides (Figures S25 and S26). Indeed, it is worth mentioning that caution is required when fixing the single-cell sample with the formaldehyde PBS solution. The fixing time, temperature, and concentration of formaldehyde should be kept at a low level to avoid excessive fixation, as the over fixing can negatively affect the detection of amine-containing lipids such as phosphatidylethanolamine (PE) and phosphatidylserine (PS).”

The dual polarity mode would also benefit from validation with standards to ensure that the steps in between and the large amount of energy put into the small space don't alter the speciation, or increase fragmentation. To ensure that there are no difference in the reanalysis of the cells, the authors should include data showing that reanalysis in positive ion mode provides the same results as the first analysis. Please include these experiments.

Response: We appreciate the reviewer for this helpful comment and excellent suggestions about validation experiments. Following the suggestion, we compared the dual-polarity ionization method and single-polarity ionization for the SC-MSI. Briefly, for the single-polarity ionization experiment, the PSC cells on the ITO slide were MS imaged under negative mode with DAN matrix directly; while for the dual-polarity ionization experiments, the PSC cells were first MS imaged under positive mode using the CHCA matrix, then the CHCA matrix were washed away with 50 mM ammonium acetate aqueous and the same PSC cells were subjected to negative mode MSI using the DAN matrix. The DAN matrix application condition was identical in both scenarios. As depicted in Figure S16, the mass spectra exhibit a high degree of consistency between the single-polarity ionized and dual-polarity ionized PSC cells though the signal intensities of some lipid species were slightly reduced in the dual-polarity ionization scenario. It's important to note that during the first run of MSI on single cells, it is crucial to control the laser energy and avoid using excessive energy to irradiate the cellular samples. Higher laser energy can potentially damage or even burn out the cells on the ITO slide, leading to decreased performance in the second MSI run. Careful control and management of laser energy is necessary to ensure optimal results in subsequent MSI run.

We understand the reviewer's concern about the first run of MSI could potentially result in the disruption of lipid species and fragmentation in the second MALDI analysis. In Figure S16 we compared the negative ion mode mass spectra obtained from PSC cells that without prior-positive-mode MSI and the mass spectra of PSC cells with dual-polarity ionization (positive MSI using the CHCA matrix, 50 mM ammonium acetate washing, followed by negative MSI with DAN matrix). The result shows the lipid species detected after the first-MSI-run and matrix-washing were quite consistent with that obtained from cells without these

pretreatments (Figure S16). The morphology of the cells was well preserved after the 1st MS imaging run, and the MS images obtained from the second MSI acquisition suggested that the extent of chemical delocalization caused by ammonium acetate washing was generally considered to be minimal (Figure S17 and Figure 3). In our experiment, the CHCA matrix was washed away using 50 mM ammonium acetate. It is noted that ammonium acetate aqueous is a widely used washing solution in MALDI-MSI. This is because ammonium acetate is a mild salt that does not react strongly with most substances, and it does not cause significant changes in the chemical properties or structure of the substance being washed. For instance, an earlier study has shown that the use of ammonium acetate (50 mM) to wash mouse brain tissue sections can result in a noteworthy improvement in signal intensity and lipidome coverage during negative ion mode MALDI-MS analysis, without any indication of delocalization at a lateral resolution of 10 μm (Richard Caprioli et al, *Anal. Chem.* 2012, 84, 3, 1557–1564).

Figure S16. Mass spectra obtained from PSC SC samples under negative mode with DAN matrix: (a) mass spectrum obtained from PSC cells undergo single-polarity ionization, (b) mass spectrum obtained from PSC cell undergo dual-polarity ionization (1st positive MSI using the CHCA matrix, followed by negative MSI with DAN matrix), (c) zoom-in mass spectra of spectra a, (c) zoom-in mass spectra of spectra b.

Figure S17. sequential MS imaging of the same single cells using dual-polarity ionization strategy: (a) bright-field image of the PSC cells on the ITO slide prior to MS imaging, (b) bright-field image of the PSC cells after the positive ion mode MS imaging (the cells were coated with CHCA matrix and the dash rectangle indicates the MSI area), (c) microscope optical image of the PSC cells after the CHCA matrix washed away using 50 mM ammonium acetate, (d)-(g) representative MS images of the PSC cells obtained via the positive mode ionization SC-MSI, (h)-(k) representative MS images of the same PSC cells obtained from the subsequent negative ion mode ionization SC-MSI. All SC-MSI images were obtained with mass error tolerance of 10 ppm. Scale bar, 400 μ m.

Following the reviewer's suggestion and comment, we added the additional experimental results and discussions into the revised manuscript, as below:

"As shown in Figure S16, the mass spectra exhibit a high degree of consistency between the single-polarity ionized and dual-polarity ionized PSC cells, though the signal intensities of some lipid species were slightly reduced in the dual-polarity ionization scenario. Further, the MS images obtained from the second MSI suggested that the delocalization of the lipid species was relatively minimal (Figure S17 and Figure 4)."

"It is important to note that, in dual ionization of single cells, it is crucial to carefully control the laser energy at an appropriate level and refrain from using excessive laser energy during the first MSI run, as excessive laser energy could damage the cells on the ITO slide and ultimately compromising the performance in the subsequent MSI run. In our experiment, the CHCA matrix was washed away with 50 mM ammonium acetate after the first MSI run. The extent of lipids delocalization caused by ammonium acetate washing is generally considered to be minimal. This has been confirmed by an earlier study that the use of ammonium acetate to wash mouse brain tissue sections can result in a noted improvement in signal intensity and lipidome coverage during negative ion mode MALDI-MS analysis, without any indication of delocalization at a lateral resolution of 10 μm^2 ."

There is a difference between mass accuracy and mass resolution. The TIMS-TOF instrument has a modest resolution, stated at 20 000 at m/z 600, which does not necessarily warrant annotation by mass alone. There are a few MSMS spectra in the SI for the manuscript, but there are annotations in the SI tables that are 10 ppm away. This suggests that these may not be the actual species. Please include data confirming these annotations.

Response: We appreciate the reviewer for this helpful comment and raising this important point. We agree with the reviewer that the TIMS-TOF instrument has a modest resolution which is not enough for the peak annotation based on the mass alone. Thus, apart from *in situ* MALDI-MS/MS spotting on individual single cells on the ITO slide, we analyzed the cellular lipid extracts via the HPLC-nanoESI-MS/MS on an Orbitrap MS platform, which provides higher resolution and more accurate mass information for further data validation.

In addition, the cellular lipid extracts were further subjected to HPLC-nanoESI-MS/MS on the TIMS-TOF platform. Thus, the ion mobility information of the cellular lipids was further collected to help the lipid identity assignment. The CCS information of the lipids has been added into the revised Table S1 and Table S2.

Additionally, it seems like the authors are assuming that they have only protonated species present in the mass spectra. However, there are some masses that overlap between protonated and sodiated species, where the sodiated annotation would make more sense. I advise that the authors combine the use of standards and endogenous species and combine this with their trapped ion mobility feature and MSMS to ensure that the annotations are indeed accurate and that there are no sodiated or potassiated peaks in the mass spectra. For example, the ion at 782.5685 is more likely a sodiated PC 34:1.

Response: We appreciate the reviewer for this comment. It could be possible to have both protonated and sodiated isobars in a SC-MALDI mass spectrum. However, we assigned mainly protonated species based on three reasons as following:

1) During the sample preparation of SC-MALDI MSI, we seeded both types of cells on the ITO slide, which we then washed with ammonium acetate aqueous solution (50 mM) for three times to mitigate the metal adducts. (See “Sample preparation for single-cell MS imaging” in Method.) It has been reported that ammonium acetate aqueous solution can help desalt the samples for lipidomic analyses and generate mainly protonated species (Wang, H. Y., Liu, C. B., & Wu, H. W. (2011). A simple desalting method for direct MALDI mass spectrometry profiling of tissue lipids. *Journal of lipid research*, 52(4), 840–849. <https://doi.org/10.1194/jlr.D013060>)

2) In Figure S13d, we performed the on-tissue MALDI-MS/MS for m/z 782.6, which we tentatively annotated as $[\text{PC } 36:4 + \text{H}]^+$. The result showed that the fragmentation of m/z 782.6 yielded a high abundance of phosphocholine fragment (m/z 184), which generally considered as the signature fragment to identify protonated PC species.

3) We performed the LC-TimsPASEF-MS/MS analysis for cell lipid extracts. And we annotated m/z 782.5731 to $[\text{PC } 36:4 + \text{H}]^+$ with a delta m/z of 4.7 ppm and delta CCS value of -0.9%. The two-dimensional data can further confirm the identities of lipid species.

How do the authors know that the differences in PL species are because of the cells and not the methodology? A cell can grow more flat or more compact on the surface. Depending on the penetration depth of the laser, it is safe to assume that lipids sampled from a more compact cell would have less influence of plasma membrane lipids than lipids sampled from a cell that is flat on the surface. Please compare cells of the same size and structure to show that the differences are indeed from biological differences and not methodological.

Response: We thank the reviewer for this great point! We agree that the height or depth of individual cells can vary across their surface, providing a meaningful analytical challenge for the single-cell MS imaging. This is why subcellular resolution is so important.

Indeed, comparing cells of the same size and structure is a great suggestion for exploring the biological differences and determine whether the differences of lipids come from the cells itself or caused by variations of cellular morphology; however, molecular heterogeneity still exists across individual cells. This means that it is difficult to prepare identical single cell samples with the same size, structure, and chemical components in current study. Please note that: 1) for the SC-MS imaging, the MALDI matrix was coated on the cells via sublimation to ensure that the matrix was evenly coated on the cells, which reduced the variations caused by the MALDI matrix; 2) for the single-cell data analysis, the statistical analysis was based on weighted average mass spectra from each individual cell region, rather than using the mass spectrum from each pixel. We believe that this approach would help to reduce the impact of cell morphology variance to some extent. Nevertheless, we will continue to work on these issues in our future research. Regarding this point, we added more discussion about this issue in the Discussion, as below:

“It is worth mentioning that the variation in height or depth of individual cells across their surface poses a notable analytical challenge for the SC-MS imaging. SC-MSI would be enhanced by improved spatial resolution and SC sectioning strategy to reduce the impact of potential variations resulting from changes of cell morphology.”

Before analysis the cells are exposed to so many different treatments. If the authors want to find biologically relevant data, would it not be better to analyze cells in their native state? Please elaborate.

Response: We appreciate this great point! We completely agree that analyzing single cells in their native state can provide the most accurate information about cellular biological processes and can offer more relevant data for downstream analysis. Undoubtedly, we strongly believe that performing native analysis of single cells would be a significant future direction for single-cell MS imaging. However, current MALDI MS-based single-cell imaging methodologies are restricted to sample pretreatments, such as buffer washings, or require vacuum conditions during ionization.

Regarding this, we add more discussion about this issue in the Discussion, as below:

“We noted that current MS-based single-cell imaging methodologies often involve sample pretreatments, such as buffer washings, or require vacuum conditions during ionization, which may disrupt the native status of the cells and lead to inaccurate biological information. To address these limitations, we believe that a promising future direction of single-cell MS imaging is to perform native analysis of single cells, as analyzing the single cells in their native state can provide more precise information on cellular biological processes and offer more relevant data for downstream analysis.”

Apart from the dual polarity analysis and different cells, it is not clear how this methodology differ from imaging of tissue with high spatial resolution (small laser spot size) MALDI and single cell studies with MALDI (including the SpaceM mentioned in the manuscript) that has been published previously (For example DOI 10.1007/s00418-013-1097-6, DOI: 10.1007/s13361-014-0883-2, DOI:10.1038/s41592-021-01198-0). Please specify.

Response: We thank the reviewer for pointing out this issue and we sincerely appreciate this suggestion. Previous publications that used high spatial resolution MALDI-MSI for single-cell analysis have been cited and discussed in the introduction part. In this study, we focus on single cell lipidomic analysis specifically and we introduced several novel aspects including:

1) We developed and implemented a **dual-ionization strategy for single-cell MS imaging**, which can improve the lipidome coverage and throughput in single cell analysis, greatly enhanced the information that can be extracted from the same single cells.

2) We incorporated **ion mobility separation as an additional dimension of analysis with distinct advantage in the single-cell MS imaging**. By incorporating ion mobility separation into MS analysis, more effective separation and identification of complex biomolecules can be achieved that are difficult to distinguish using MS alone. The use of ion mobility in single-cell MS imaging has not been demonstrated.

3) The revised manuscript includes **a novel developed graphical user interface (GUI) that provides a user-friendly platform for analyzing single-cell MSI data**. This GUI offers more streamlined and automated single-cell MSI data analysis, allowing single-cell mass spectra picking and statistical analysis. We have included a description of the automated ROI selection process in the revised manuscript.

4) As a proof-of-concept study, we introduced **in-situ single-cell epoxidation chemical derivatization** for localization of carbon-carbon double bond and in-depth analysis of lipid C=C isomers at the single-cell level.

Taken together, we believe that our study is distinctly different from previous research endeavors in this rapidly expanding research field where new sample preparation methods and improved data analysis strategy and unique applications are needed.

Minor comments

In Figure 5 the authors detail how the implementation of ion-mobility improved the separation of isobaric compounds 720.5497 and 720.5866 m/z and resulted in dramatically different distributions of these two lipids between PANC1 and PSC cells. However, as can be seen in Figure 5 a) the two ions can be sufficiently separated by MS as well. Have the authors tried to depict the spatial distribution of these two lipid species based on solely the m/z data? It would be interesting to see a comparison between the UMAP plots of PANC1 and PSC cell subpopulations with and without using CCS data.

Response: We appreciate the reviewer for raising this interesting point.

1) Following this helpful suggestion, new data has been added to the revised manuscript to demonstrate the advantage of ion mobility separation for lipid isobars and isomers.

Figure 3. SC-MSI of the PSC cells coupled with the ion mobility separation: (a) representative ion-mobility MS data collected at a narrow mass window of *m/z* 762.5 to 762.8 Da, (b) MS image constructed based on the *m/z* value only, (c-f) MS image of isobaric and isomeric lipids revealed based on the combination of their *m/z* value and the collisional cross sectional (CCS) information. Scale bar, 400 μm .

A description of this new data has been added to the revised manuscript accordingly, as below:

“We also noticed that more lipid species could be revealed from SC with the assistance of ion mobility separation, as illustrated in Figure 3, while only one peak was observed when using MS alone, the incorporation of ion mobility separation enabled the resolution and detection of four distinct lipid species. In addition, variations in lipid isobaric/isomeric compositions between the PSC and PANC1 cells were observed with ion mobility separation (Figures S10 and S11). These results demonstrate that ion mobility provides a distinct advantage in revealing isobaric and isomeric lipid species in SC-MSI.”

Furthermore, we also added additional TIMS data into the Supporting Information, showing the lipid difference between the PSC and PANC1 cells revealed by SC-MALDI-MSI coupled with ion mobility separation. The data are shown as below:

Figure S10. Average ion mobility heat maps and mass spectra obtained from the PSC cells (a) and PANC1 cells (b) using the SC-MALDI-MSI strategy coupled with ion mobility separation under positive ionization mode. The heat map displays a diverse range of ions for each m/z value, showing an enhanced peak capacity achieved through the integration of the TIMS separation dimension. The mass spectra ranges include m/z values 700–850 and the heat map incorporates $1/K_0$ values 1.35–1.49.

Figure S11. Representative zoom-in ion mobility heat maps and mass spectra obtained from the PSC cells and PANC1 cells using SC-MALDI-MSI coupled with the ion mobility separation under positive ionization mode. (a) PSC cells at the mass spectral range of m/z 762.4–762.8, (b) PANC1 cells at the mass spectrum

range of m/z range 762.4–762.8, (c) PSC cells at the mass spectral range of m/z at 788.4–788.8, and (d) PANC1 cells at the mass spectral range of m/z 788.4–788.8.

2) We thank the reviewer for this great point of including the CCS data in the UMAP analysis of PANC1 and PSC cell subpopulations! The current UMAP analysis is based on the average mass spectral data extracted from each single-cell region and a full discrimination of the PANC1 and PSC cells was achieved using the UMAP and machine learning analysis. Indeed, we strongly agree that adding the CCS information into the analysis could enhance data analysis. Unfortunately, we encountered a limitation in that the CCS data from each individual cell region cannot be exported separately, which currently restricts the integration of CCS data into our analysis. We also brought this issue to the attention of Bruker application team, who informed us that it is not possible to extract CCS data from separate regions in the TIMS-MALDI-MS imaging platform currently, but they are working on developing this function for future versions. We think this would be an important future direction in our follow-up study.

Regarding this, we added more discussion about this issue in the Discussion, as below:

“It is noted that the current statistical analysis of single-cell data is based on the average mass spectral data extracted from each single-cell but does not include the CCS data. We believe that adding the CCS information into the analysis pipeline could greatly enhance the information extracted from these single cells. Unfortunately, we encountered a limitation in that the CCS data from each individual cell region cannot be exported separately, which currently restricts the integration of CCS data into our analysis. We believe that integrating CCS data into our analysis will be an important future direction for our study.”

Do the authors have any data on CCS values of structural isomers or were they indistinguishable by ion-mobility MS?

Response: We thank the reviewer for this question. We have included additional TIMS data into the revised manuscript to demonstrate the advantages of ion mobility separation for lipid isobars and isomers (Figure 3, Figure S10, Figure S11). Additionally, we have incorporated the CCS information of the lipids into the revised Table S1 and Table S2.

I generally miss the information on spatial resolution for the different single-cell images. Some images seem to have higher resolution than others, for example, images in Figures 2 and 3 seem to have higher spatial resolution than those in Figure 4. It would be nice to include details on spatial resolution either in the text or in the image descriptions, as the authors did in the case of brain tissues. I also didn't find any data about cell sizes which would be useful to include as well to get a deeper insight into the extent of subcellular resolution.

Response: We thank the reviewer for this helpful suggestion. Details about the spatial resolution have been added into the image descriptions. All the single-cell samples were imaged and analyzed at 10 μm spatial resolution. PANC1 and PSC cells have different shapes and sizes on the ITO slide, which can be easily differentiated by their distinct morphology. PANC1 cells in culture typically appear as round epithelial cancer cells with a diameter ranging from 18-25 μm . On the other hand, PSC cells have an elongated fibroblast-like morphology with protrusions and a large length to width ratio (Figure S19). The average size of PSC cells is 114 μm x 43 μm , which is approximately 2.8 times larger than that of PANC1 cells.

Information about the cell sizes has been added into the revised manuscript in Figure S19.

In Supplementary Figure S4 when comparing the MALDI matrix depositing techniques, the authors are using different magnification values in each case, which makes them hardly comparable. Please adjust to the same magnification.

Response: We thank the reviewer for this helpful suggestion. We are sorry for the misleading description of Figure S4, and this issue has been addressed in the revision.

Please include a motivation to the choice of matrix and why only two matrices were tested.

Response: We thank the reviewer for this suggestion. Herein, we choose DHB and CHCA as matrix in the positive mode ionization, as they are commonly used matrices for MALDI-MS. However, it is important to note that the selection and deposition of matrix can have a significant effect on the quality of results, and this should be taken into consideration for specific types of analyses. We now included more explanation about our rationale of matrix choice in the revised manuscript, as below:

“Herein, we used DHB, CHCA, and DAN as MALDI matrices, as they are commonly used for metabolite analysis. It is important to note that the selection and deposition of matrix could have a significant effect on the quality of results, and this should be taken into consideration for specific type analyses.”

In the Figures the authors state that the displayed ion images is of lipids annotated within 5 ppm, but in the SI tables the error is much larger. How many of the 180 lipids that the authors claim to have annotated are actually within the 5 ppm? Please specify.

Response: We thank the reviewer for pointing out this. Sorry, it was typo error, the displayed ion images were constructed within 10 ppm. This typo error has been corrected in the revised manuscript.

In Figure 4 it is unclear what cells that are of which kind. Please include a specification on how the authors know what cell is PANC1 and PSC.

Response: We thank the reviewer for this helpful suggestion.

Both PANC1 and PSC cells were characterized based on their cell surface markers and protein expression by collaborators before use. PANC1 was obtained from a commercial source (ATCC), while PSC was isolated from a pancreatic cancer patient sample. These two cell types are from different lineage and have different morphology. These cells have different shapes and sizes on the ITO slide that can be easily differentiated by distinct morphology. PANC1 cells are typically round with a diameter of 18–25 μm , resembling typical cancer epithelial cells. On the other hand, PSC cells are elongated with protrusions, having larger length to width ratios (Figure S19). The average size of PSC is $114 \times 43 \mu\text{m}^2$, which is typically 2.8 times larger than that of PANC1.

Based on several parameters, such as cell length, width, areas, perimeters cell roundness and circularity, ImageJ particle analysis and MALDI imaging software can differentiate these two cell types with high confidence. In original Figure 4 (changed to Figure 5 in revision), the cells denoted by blue and red arrows in our figures were identified by software based on the mentioned parameters. For instance, PSC cells were identified as having a cell area of 300-800 μm^2 , length to width ratio greater than 1.5, and circularity PANC1 cells were identified as having a cell area less than 300 μm^2 , length to width ratio less than 1.4, and less than 0.4. PANC1 cells were identified as having a cell area less than 300 μm^2 , length to width ratio less than 1.4, and circularity between 0.5 and 1.0.

Figure S19. Both PANC1 and PSC cells were characterized based on their cell surface markers and protein expression by collaborators before use, these two cell types were from different lineage and have different morphologies. (Top) Bright field images of PANC1, PSC and PANC1+PSC in coculture. All scale bars were equal to 50 μm . (Bottom) NIH ImageJ software was used to measure cell length, width, areas and perimeters and derived length to width ratio and circularity when PANC1 and PSC cultured individually. The length to width ratio was roughly 1.2:1 for PANC1 whereas that ratio is 2.6:1 in PSC. We didn't find these parameters change when these two cells cultured together in coculture system. These cells have different shapes and sizes on the ITO slide that can be easily differentiated by morphology. PANC1 cells are typically round with a diameter of 18–25 μm , resembling typical cancer epithelial cells. On the other hand, PSC cells are elongated with protrusions, having larger length to width ratios. The average size of PSC is $114 \times 43 \mu\text{m}^2$, which is typically 2.8 times larger than that of PANC1.

A description of the coculture sample has been added into the revised manuscript as below:

“These two cell types are from different lineage and have different morphology (Figure S19).”

In Figure 6 it seems like treatment above 1 μ M has significant changes, however, the authors state above 10 μ M. Please explain your reasoning for choosing this higher number.

Response: We thank the reviewer for pointing out this error. It was a typo, and this has been changed to 10 nM. Cell viability started to decrease when MF-438 was above 1 nM but the change was not statistically significant until reaching 10 nM concentration. So, we decided to choose 10 nM for all experiments.

Additional revision:

Additional experiments with details about in-situ single-cell epoxidation chemical derivatization was added into the Supporting Information as below:

In-situ single-cell epoxidation chemical derivatization

The cell slide washing and fixing step is the same as detailed in the main text. The workflow of epoxidation chemical derivatization for single cell on the ITO slide is similar to our previous study, with slight modification (Chemical science 12 (23), 8115-8122). Briefly, the single cell slide incubated in a sealed chamber with peracetic acid (PAA) vapor for 1 h under room temperature. To generate PAA vapor, 5 mL 10% (v/v) PAA aqueous solution was placed in the bottom of the chamber; one needs to exercise caution not to allow the slide contact with the solution. Following the PAA epoxidation, the single-cell slide was retrieved from the chamber followed by MALDI matrix application via sublimation. For MALDI tandem MS analysis using the TIMS-TOF-MALDI platform, the high-energy collisional dissociation (HCD) MS/MS was performed with precursor ions isolated at a mass window width of 1 Da and 35 eV collision energy. Other MS instrumental parameters were set to default values without any further optimization. For nanoESI-MS analysis of lipids extracted from a PAA treated single-cell slide, the cells on the slide were extracted with 0.2 mL of 50% ACN/MeOH solution, and the samples were diluted 10 times using methanol before direct nanoESI-MS on an Orbitrap Fusion Lumos mass spectrometer. For tandem MS analysis on the Orbitrap platform, the HCD MS/MS was performed with precursor ions isolated at a mass window width of 0.8 Da, and a normalized collision energy (NCE) of 20%. Other MS instrumental parameters were set to default values without any further optimization. The experimental data were shown in **Figure S28** and **Figure S29**.

Additional discussion was added in the revised manuscript as below:

“For example, our group recently developed a novel approach of using peracetic acid (PAA)-induced epoxidation coupled with tandem MS for spatial interrogation of the C=C positional lipid isomers from biological samples⁵¹. Such PAA epoxidation derivatization could be coupled with SC-MSI for further enhancement of SC lipidomics, revealing additional structural isomers. In a proof-of-concept experiment, we observed that cells retained their original morphology following treatment with PAA vapor for in-situ epoxidation derivatization, and that the lipid PC(34:1) (m/z 760.5853) within the cells was successfully converted to an epoxy-PC(34:1) product (m/z 776.5803) (**Figure S28**). However, we noted that our current TimsTOF-MALDI MS platform lacks the necessary sensitivity to detect diagnostic reporter ions specific to C=C double bonds during MS/MS analysis (**Figure S29**). Despite this limitation in our current instrument, we hypothesize that in-situ single-cell epoxidation chemical derivatization could offer a promising approach for exploring lipid C=C isomers in SC-MSI analysis. Nevertheless, our ongoing efforts are directed at integration of SC-MSI with in situ chemical derivatization approaches to enhance the depth of chemical and molecular information that can be extracted during SC analysis.”

Figure S28. In-situ single-cell epoxidation chemical derivatization for in-depth analysis of lipid C=C isomers in single cells: (a) schematic diagram of peracetic acid (PAA) induced epoxidation of unsaturated lipids, in which single-cell slide sample was incubated in a chamber containing PAA vapor, the gaseous PAA reacts with the unsaturated lipids in the cells and form epoxy-lipid products; (b) bright-field image of the PANC1 cells prior to the epoxidation treatment, the bottom black line is the sharpie marker draw on the backside of the ITO slide; (c) bright-field image of the PANC1 cells after 1 h incubation in PAA vapor, the bottom black sharpie marker faded away due to the oxidation treatment, while the cells remained intact on the slide; (d) MALDI spotting mass spectrum of PANC1 cell without the PAA treatment; (e) MALDI spotting mass spectrum of PANC1 cell with the PAA epoxidation treatment. The MALDI spotting mass spectra were obtained using a 10 μm size laser irradiation on the single cell, with CHCA matrix under positive ionization mode. After the PAA treatment, we can observe that the cells remained on the ITO slide and maintained similar morphology. The PC(34:1) (m/z 760.5853) in the cells were derivatized to an epoxy-PC(34:1) product with a mass increase of 16 Da, leading to an m/z value of 776.5803. The data indicates that the PAA vapor was able to effectively derivatize unsaturated lipids in single cells, which presents a promising approach to distinguishing between different lipid C=C isomers at the single-cell level through tandem MS.

Figure S29. MS/MS spectra of precursor ion of epoxy-PC(34:1) (m/z 776): (a) HCD MS/MS spectra of precursor ion m/z 776 via MALDI spotting on the PAA treated single-cell sample using the TIMS-TOF-MALDI platform; (b) nanoESI-MS/MS (HCD) of precursor ion m/z 776 via direct infusion of lipids extracted from a single-cell slide treated with PAA, using a Fusion Lumos Orbitrap MS platform; (c) zoom-in mass spectra at a mass window of m/z 450-800 from b. The results showed that the TIMS-TOF MS lacks sufficient sensitivity to detect C=C double bond diagnostic reporter ions, while detected abundant phosphate head group of the epoxy-PC(34:1). However, two pairs of diagnostic ions m/z 634.44 and 650.44 and m/z 662.47 and 678.47 were detected using the nanoESI-MS/MS on an Orbitrap MS platform, indicating there are two lipid C=C double bond isomers of PC 16:0_18:1(9) and PC 16:0_18:1(7).

REVIEWERS' COMMENTS

Reviewer #1 (Remarks to the Author):

The authors have fully addressed my comments. The current version is ready for publication. Congratulations!

Reviewer #2 (Remarks to the Author):

Zhang et al. present a thorough review of their manuscript and have replied to all questions raised by the reviewers. They have added some additional experiments and present an amended data analysis strategy. As requested by all three reviewers, the authors emphasize the novelty of the presented work and try to distinguish it more clearly from existing work. In my view, this was only partly successful. Instead of exploiting the novel aspects of adding dual polarity and IMS capabilities to SC analysis to the full extend, they add new aspects of analysis. While these new aspects were added, very little content was cut or shortened. This makes the paper rather lengthy and in parts repetitive.

Detailed comments:

Results:

Page 5 lines 92ff: In their rebuttal and also in the methods section, the authors describe how sample preparation was largely inspired by previously published work. The comparison of sublimation and spray based preparation as well as fixation and washing are therefore for the most part not novel results. They should be shortened and moved to the methods section. A more detailed description about the optimization of laser parameters to allow for dual polarity measurements would be much more informative. How are critical parameters like laser pulse energy, number of laser pulses and laser repetition rate evaluated? If a more volatile matrix such as 2,5 DHA would be used, would sublimating the first matrix off the sample be a feasible alternative to washing?

Page 6 line 131: Could you describe (in the methods) how co-registration between optical and MS images was executed and how you measure the fidelity of co-registration? Are images in positive and negative ion mode co-registered independently? Do you record a new microscopic image after washing of the first matrix?

Page 7 figure 1: The schematic workflow does not include the newly added software based identification of cells. It would be interesting to see where that fits in.

Page 11 and other places: Throughout the manuscript you use 50 mM ammonium acetate washes. Could you explain why you use 50 mM? Wouldn't an iso-osmotic concentration of ~150 mM be better suited to conserve the morphology and integrity of the cells?

Page 12 figure 4: The washing step is missing from the schematic diagram

Page 13 line 222 ff: I assume that all presented results are based on the "old" work-flow already presented in the first version of the manuscript it would be interesting to see how results based on the new automated workflow would compare.

Page 14 line 242 and also figure 6: In this part of the manuscript, it remains unclear that the additional information gained by IMS is not used in the UMAP but that it is solely based on m/z values. It is only clarified much later in the paper but should also be mentioned at this point. It is also unclear if combined data of both polarities is used for the analysis of heterogeneity and statistical analysis. As both IMS and dual polarity are the most novel and most interesting aspects of this work, this should be described more carefully and with all necessary detail. It would, for example be interesting to see how much better the combined data performs as compared to positive or negative ion mode alone. How much better does it perform with IMS data added (also see comment below). Can you somehow quantify the gain in information compared to the status quo?

Page 14 line 250: If I understand correctly, all cells included in the machine learning experiment are "hand picked". Does this include all cells within a certain area of the slide, or where cells that may have clustered or that are smaller than usual excluded? For a fair comparison, all cells within a certain area should be considered regardless of their size and shape.

Page 17 line 283: How do you differentiate free fatty acids from fragments that may be produced during the MALDI analysis?

Page 19: The whole chapter about the analysis of brain tissue does contain very little novel aspects. Segmentation of MSI data at 10 μm resolving power and direct comparison to anatomical features has been shown before. It also does not fit well with the "single cell" theme of the paper. I would suggest therefore suggest to concentrate on single cell analysis from cell culture.

Discussion: The discussion part is rather long. It should be streamlined and checked for repetitions and redundancies.

Page 24 line 410: The part of the discussion describing the structural elucidation of C=C positional isomers should be omitted from the paper. While it is certainly an interesting approach and was demonstrated for tissue, the authors could not demonstrate a proof-of-principle on single cells due to lack of signal intensity in MSMS spectra. The mere pointer that it works for tissue and therefore should also work for cells is not valid. If it were, this would also challenge the novelty of IMS and dual polarity measurements for SC as both techniques have also been demonstrated on tissue before.

Page 25 line 438: The described GUI and software based extraction of single cell mass spectra is sort of disconnected from the rest of the paper. It remains unclear where the authors have used the new technique. Being based solely on MS data, it significantly differs from the microscopy based strategy used earlier by the authors of this paper as well as other. It would therefore be interesting to see how it compares to the more established methods. This would be especially interesting on denser cell populations where cell clusters and touching cells are more the rule than the exception. Wouldn't most of these clusters be excluded from the analysis? And would that not introduce a biologically relevant bias towards isolated cells as compared to cells in contact with other? Also for the co-culture of cells of different size, how strict do the rules have to be to be able to detect all cells of both types without false positives or negatives?

Methods:

Page 29 line 524: There seems to be a typo: What do you mean by a pressure of 350 microns?

Page 31 line 561: It remains unclear where the "old" manual and where the new automated work-flows were used. A comparison of both work-flows on the same data set would be interesting.

Page 31 line 561: How did you export mass spectra from ROIs? How did you perform peak picking for data base search?

Page 32 line 577: All cells for this analysis seem to be "hand picked" to clearly resemble one group or the other. What about cells on the sample that were not distinctly identified? How do results compare if cell picking is done with the new automated software tools?

Page 32 line 588: The information that statistical analysis does not include the IMS dimension needs to be clarified in the results and not only in the methods. The claim, that CCS data cannot be exported separately from ROIs is not correct. To my knowledge, SciLSLAB API provided by Bruker allows to extract feature lists with ion mobility data from ROIs. Also independent software such as pyxis by mass analytica allows to process IMS-MS imaging data produced with timsTOF flex instruments.

REVIEWERS' COMMENTS

Reviewer #1 (Remarks to the Author):

The authors have fully addressed my comments. The current version is ready for publication. Congratulations!

Response: We are delighted that the reviewer acknowledges our efforts in addressing all the comments thoroughly and think our current version is acceptable for publication. We appreciate the reviewer's critical and helpful comments to help improve our manuscript.

Reviewer #2 (Remarks to the Author):

Zhang et al. present a thorough review of their manuscript and have replied to all questions raised by the reviewers. They have added some additional experiments and present an amended data analysis strategy. As requested by all three reviewers, the authors emphasize the novelty of the presented work and try to distinguish it more clearly from existing work. In my view, this was only partly successful. Instead of exploiting the novel aspects of adding dual polarity and IMS capabilities to SC analysis to the full extend, they add new aspects of analysis. While these new aspects were added, very little content was cut or shortened. This makes the paper rather lengthy and in parts repetitive.

Response: We appreciate the reviewer's generous help and constructive comments on our manuscript. We have revised the manuscript following the reviewer's comments, including 1) cutting and shortening some lengthy content to make the manuscript more concise, and 2) reanalyzing the single-cell MS imaging data to exploit the advantages of dual-polarity ionization and ion mobility separation. Following the constructive suggestions from the reviewer, we think the quality of the manuscript has been significantly enhanced.

Detailed comments:

Results:

Page 5 lines 92ff: In their rebuttal and also in the methods section, the authors describe how sample preparation was largely inspired by previously published work. The comparison of sublimation and spray based preparation as well as fixation and washing are therefore for the most part not novel results. They should be shortened and moved to the methods section. A more detailed description about the optimization of laser parameters to allow for dual polarity measurements would be much more informative. How are critical parameters like laser pulse energy, number of laser pulses and lase repetition rate evaluated? If a more volatile matrix such as 2,5 DHA would be used, would sublimating the first matrix off the sample be a feasible alternative to washing?

Response: We thank the reviewer for this helpful comment.

1) In the Results section, the content about the comparison of sublimation and spray based preparation, as well as fixation and washing, have been shortened and moved to the Method section.

2) A detailed description of the laser parameters to allow for dual polarity measurements has been added into the Method section, as below:

“Specifically, in dual-ionization MS imaging of SCs, laser-shots of 50 (5000 Hz) were used with relatively lower laser energy ca. 25% in the first positive MSI run using CHCA as matrix; for the subsequent negative imaging run, the 100 shots (5000 Hz) with laser energy of 40% were conducted, using DAN as the MALDI matrix. It is important to note that, in dual ionization of single cells, it is crucial to carefully control the laser energy at an appropriate level and refrain from using excessive laser energy during the first MSI run, as excessive laser energy could damage the cells on the ITO slide and ultimately compromise the performance in the subsequent MSI run.”

3) We thank the reviewer for this great suggestion! We agree that for more volatile matrix, sublimating would be an alternative approach to remove the first matrix off the samples. In this study, we used CHCA matrix in the first MS imaging run, which is relatively vacuum stable compared to 2,5 DHA. Therefore, we used ammonium acetate solution for washing.

Page 6 line 131: Could you describe (in the methods) how co-registration between optical and MS images was executed and how you measure the fidelity of co-registration? Are images in positive and negative ion mode co-registered independently? Do you record a new microscopic image after washing of the first matrix?

Response: We thank the reviewer for this comment. Detailed description of the co-registration between the optical and the MS images has been added into the Methods. We admit that currently the co-registration of the optical and MS images was manually performed when comparing the optical and MSI images of the single cells. We apologize for the misleading description, and the original language, “, which demonstrated a high-fidelity of coregistration between the microscopic images and the mass spectrometric fingerprints of each individual cells”, has been removed.

Images in positive and negative ion mode were compared to their optical images independently during data analysis. After the matrix washing and drying, an optical image of single-cell slide was recorded prior to applying another MALDI matrix sequentially (as shown in **Supplementary Figure 17**). These details have been added into the Methods.

The descriptions below have been added into the Method in revised manuscript:

“Coregistration of MSI images and the optical image of the cells on the slide were performed by combining the cell microscopic image and the MSI image.”

“After the matrix washing and drying, an optical image of single-cell slide was recorded again prior to applying a MALDI matrix sequentially, and the ion images in positive and negative ion mode compared to their optical images independently during data analysis.”

Page 7 figure 1: The schematic workflow does not include the newly added software based identification of cells. It would be interesting to see where that fits in.

Response: We thank the reviewer for this important comment. We have added the software information into revised the schematic workflow accordingly. Revised Figure 1 as below:

Figure 1 Schematic workflow of SC analysis based on the high spatial resolution MALDI MS imaging. For details about the experimental workflow refer to the Method section.

Page 11 and other places: Throughout the manuscript you use 50 mM ammonium acetate washes. Could you explain why you use 50 mM? Wouldn't an iso-osmotic concentration of ~150 mM be better suited to conserve the morphology and integrity of the cells?

Response: We thank the reviewer for this insightful comment. A 50 mM ammonium acetate wash was used based on a previous study (Anal. Chem. 2012, 84, 3, 1557–1564). We find that the cell morphology is largely conserved by the formalin fixation step before the ammonium acetate solution washing. As pointed out by the reviewer, a 150 mM concentration of ammonium acetate could be an alternative buffer for cell washing. In our experience, we observed that residual of ammonium acetate remains on the surface of the cells after the washing process, and its concentration increases significantly prior to crystallization during the drying process. To reduce the potential impact of ammonium acetate crystals on single-cell MS imaging, we have opted to use a washing solution containing 50 mM ammonium acetate. However, conducting a systematic comparison of different concentrations of ammonium acetate solutions would yield more valuable information. This aspect will be addressed in our future studies.

Page 12 figure 4: The washing step is missing from the schematic diagram.

Response: We thank the reviewer for this constructive comment. The washing step has now been added into Figure 4 in the revised manuscript. Revised Figure 1 as below:

Figure 4. Multimodal SC-MSI of individual cells.

Page 13 line 222 ff: I assume that all presented results are based on the “old” work-flow already presented in the first version of the manuscript it would be interesting to see how results based on the new automated workflow would compare.

Response: We thank the reviewer for this great comment. Following this suggestion, we added the results based on the new automated workflow into the Supporting Information. As shown in **Supplementary Figure 31**, and **Supplementary Figure 32**, for the analysis a PANC1 cell MSI data set, 87 PANC1 cells were tentatively picked with SCiLS Lab manually (**Supplementary Figure 31c**, some of PANC1 cell clusters were omitted during the ROI picking) based on the comparison of optical ion images. The proposed automated workflow identified 198 cells (**Supplementary Figure 31d**). The automated selected ROIs successfully include all the manually picked ROIs. The implementation of the automated workflow has considerably improved the efficiency of data processing, with which the ROI picking can be completed within minutes, whereas the manual picking via SCiLS Lab would typically take several hours.

The results indicate that the automated workflow provides user friendly way for the analysis of single-cell MSI data, however, optimization and iteration of the current automated program is still required. Our strategy in the design of this new automated workflow was to have it serve as a complement to our existing manual segmentation strategy. We achieved this by producing a program capable of extracting single cells based on specific filtering. With this, the user is then able to look at the optical image with the segmented cell image in tandem to identify any cells lacking extraction. In the future, we plan to optimize this program

for more accurate extraction of cell boundaries, especially for the overlapped cell clusters (currently the overlapped cells were discarded during the ROIs picking). Previous study has shown the feasibility of recognizing the cell boundary based on a combination of cell fluorescent images with the assistance of the well-established cell segmentation platform, CellProfiler (<https://cellprofiler.org/>) (*PNAS*, 119, e2114365119 (2022)). These advances in single cell recognition have inspired a future direction for us to improve the current program in our coming work.

Supplementary Figure 31. Single-cell ROI picking of PANC1 cell MSI data using the SCiLS Lab manually and the MSI Parser automated workflow: (a) microscopic brightfield image of the PANC1 cell prior to the MS imaging; (b) ion image of PC (34:1) (m/z 760.58) from the PANC1 cells; (c) single-cell ROIs picked using the SCiLS Lab manually, in which 87 PANC1 cells were tentatively picked based on comparison of the optical image and the ion images. Here, each circle presents a single cell region, where some of PANC1 cells may be missed during the manual ROI picking process; (d) single-cell ROIs picked using the proposed MSI Parser automated workflow, in which 198 cells were picked, where each color-dot presents a single cell region. The result obtained from the automated workflow successfully included all of the manually-picked ROIs. The implementation of the automated workflow has considerably improved the efficiency of data

processing, with which the ROI picking can be completed within minutes, whereas the manual picking via SCI LS Lab would typically take several hours.

Supplementary Figure 32. Output results from automated workflow for the analysis of MSI data from PANC1 and PSC cells: (a) PCA analysis result, (b) UMAP analysis result, (c) t-SNE analysis result, (d) machine learning confusion matrix displays the result of the testing data using the Random Forest model. In the PCA, UMAP, and t-SNE results, one dot presents a single cell. MSI data from PANC1 and PSC cells were imported in the MSI Parser automated workflow separately, in which 48 PSC cells and 198 PANC1 cells were extracted from the PSC and PANC1 MSI data, respectively. These results demonstrate that the PSC and PANC1 cells could be successfully distinguished under the statistical analysis based on the automated workflow.

Page 14 line 242 and also figure 6: In this part of the manuscript, it remains unclear that the additional information gained by IMS is not used in the UMAP but that it is solely based on m/z values. It is only clarified much later in the paper but should also be mentioned at this point. It is also unclear if combined data of both polarities is used for the analysis of heterogeneity and statistical analysis. As both IMS and dual polarity are the most novel and most interesting aspects of this work, this should be described more carefully and with all necessary detail. It would, for example be interesting to see how much better the combined data performs as compared to positive or negative ion mode alone. How much better does it perform with IMS data added (also see comment below). Can you somehow quantify the gain in information compared to the status quo?

Response: We appreciate the reviewer for this very constructive comment! We agree that involving the dual-polarity and ion mobility data into the statistical analysis could provide more information about the single-cell heterogeneity. This information gain has been demonstrated in the MSI images as shown in **Figure 3** and **Figure 4**. Following the reviewer's suggestion, we re-analyzed the data from the dual-polarity ionization MS imaging experiment (the ion images showed in **Supplementary Figure 17**). The MSI data from positive mode MSI, negative mode MSI, and positive and negative combined data for each single cell were subjected to PCA and UMAP analysis. The result has been added into the Supporting Information as **Supplementary Figure 19**. As expected, different PCA and UMAP results were observed using different data sets. Briefly, the overall average mass spectra from 22 PSC cells were extracted from both the positive MSI and negative MSI data sets. Under a condition of no data intensity filtering of the raw mass spectra, 29890 m/z features from the positive MSI data, and 21563 m/z features from the negative MSI data were respectively extracted for the PCA and UMAP analysis. Therefore, a total of 51453 m/z features from each single cell were used for the statistical analysis after a combination of the positive and negative mass spectra data for each cell accordingly.

Supplementary Figure 19. PCA and UMAP analysis of the PSC cell MSI data show in Supplementary Figure 17: (a) PCA result of the PSC cells imaged under positive mode acquisition, (b) PCA result of the PSC cells imaged under negative mode acquisition, (c) PCA result of the PSC cells with positive and negative data combined for each cell accordingly, (d) UMAP result of the PSC cells imaged under positive mode acquisition, (e) UMAP result of the PSC cells imaged under negative mode acquisition, (f) UMAP result of the PSC cells with positive and negative data combined for each cell accordingly. Each dot presents a PSC cell. For the PAC and UMAP analysis, briefly, the overall average mass spectra from 22 PSC cells were extracted from both the positive MSI and negative MSI data sets. Under the condition of no data intensity filtering of the raw mass spectra, for each single cell, 29890 m/z features were extracted from the positive MSI data, and 21563 m/z features were extracted from the negative MSI data for the analysis. After a

combination of the positive and negative mass spectra for each cell accordingly, a total of 51453 m/z features from each single cell were used for the analysis.

Again, we appreciate the reviewer's suggestion about inputting the ion mobility data for the statistical analysis. Following the suggestion, we upgraded our SCiLS lab to API version and tried to extract the mass spectra data along with its CCS data. The CCS data were successfully extracted using the SCiLS lab API. As an example, we analyzed the MSI data sets from PANC1 and PSC cells using PCA and UMAP both with and without the inclusion of CCS information. This result has been added to the Supporting Information as **Supplementary Figure 20**. A total of 37 PSC cells and 43 PANC1 cells were included in the analysis. The successful separation of PSC and PANC1 cells is demonstrated in the PCA and UMAP analysis, as depicted below. Upon comparing the results obtained from the analysis using only the mass spectra data to those incorporating the ion mobility CCS information, it was observed that the inclusion of CCS data led to an improved performance in grouping the cells within the PCA and UMAP results.

Supplementary Figure 20. PCA and UMAP analysis of the MSI data from the PSC and PANC1 cells: (a) PCA result based on the mass spectra, (b) UMAP result based on the mass spectra data, (c) PCA result based on the mass spectra and CCS information, (d) UMAP result based on the mass spectra and CCS information. Here, each color-dot represents a single cell, a total of 37 PSC cells and 43 PANC1 cells were included in the analysis. The successful separation of PSC and PANC1 cells is demonstrated in the PCA and UMAP analysis. Upon comparing the results obtained from the analysis using only the mass spectra data (a and b) to those incorporating the CCS information (c and d), it was observed that the inclusion of CCS data led to an improved performance in grouping the cells within the PCA and UMAP results.

The description below has been added into the revised manuscript.

“It is worth noting that the analysis of SC heterogeneity can incorporate both positive and negative ion mode mass spectra (**Supplementary Figure 19**), owing to the dual-polarity ionization technique employed in the analysis of a single cell. This dual-polarity ionization enables a more comprehensive assessment of the molecular composition and diversity within individual cells. Furthermore, as shown in **Supplementary Figure 20**, the integration of ion mobility separation adds an additional dimension of molecular information to the profiling of SC heterogeneity.”

Page 14 line 250: If I understand correctly, all cells included in the machine learning experiment are “hand picked”. Does this include all cells within a certain area of the slide, or where cells that may have clustered or that are smaller than usual excluded? For a fair comparison, all cells within a certain area should be considered regardless of their size and shape.

Response: We thank the reviewer for this comment. We can confirm that all cells within the MSI area were picked for the data processing shown in Page 14 line 250, regardless of their size and shape. We picked the cellular ROI according to the optical image of the cells.

Page 17 line 283: How do you differentiate free fatty acids from fragments that may be produced during the MALDI analysis?

Response: We thank the reviewer for this great question. We agree with the reviewer that fragment interference from other lipid species could be a potential issue during the detection of the free fatty acids from the samples. In our experiment, we carefully controlled the laser energy at an appropriate level and refrained from using excessive laser energy during the MSI experiments, as this would minimize the interference from the lipid fragmentation. To monitor fragmentation interference more accurately, spiking odd chain or isotopic internal phospholipid standards into the MALDI matrix would be a feasible way to evaluate in-source fragmentation issue during the MALDI analysis. In this study, our primary focus is on establishing the single-cell MS imaging platform. However, we recognize the importance of conducting this follow-up experiment in our future investigations.

Page 19: The whole chapter about the analysis of brain tissue does contain very little novel aspects. Segmentation of MSI data at 10 μm resolving power and direct comparison to anatomical features has been shown before. It also does not fit well with the “single cell” theme of the paper. I would suggest therefore suggest to concentrate on single cell analysis from cell culture.

Response: We thank the reviewer for this constructive comment. Following the reviewer’s comment, we have trimmed brain tissue imaging chapter from the main manuscript and put some of the content into the Supporting Information. As a result, the revised manuscript focuses on the single cell analysis from cell culture.

Discussion: The discussion part is rather long. It should be streamlined and checked for repetitions and redundancies.

Response: We thank the reviewer for this helpful comment. We have checked the Discussion part and removed the repetitive and redundant parts. To make the discussion more concise, the contents including the description of the structural elucidation of C=C positional isomers via PAA epoxidation and the detailed discussion of MSI of mouse brain tissues have been removed.

Page 24 line 410: The part of the discussion describing the structural elucidation of C=C positional isomers should be omitted from the paper. While it is certainly an interesting approach and was demonstrated for tissue, the authors could not demonstrate a proof-of-principle on single cells due to lack of signal intensity in MSMS spectra. The mere pointer that it works for tissue and therefore should also work for cells is not valid. If it were, this would also challenge the novelty of IMS and dual polarity measurements for SC as both techniques have also been demonstrated on tissue before.

Response: We thank the reviewer for this constructive comment. Following the reviewer's suggestion, we have removed the discussion of structural elucidation of C=C positional isomers in single cell MSI and the relevant data (figures) have been removed as well.

Page 25 line 438: The described GUI and software based extraction of single cell mass spectra is sort of disconnected from the rest of the paper. It remains unclear where the authors have used the new technique. Being based solely on MS data, it significantly differs from the microscopy based strategy used earlier by the authors of this paper as well as other. It would therefore be interesting to see how it compares to the more established methods. This would be especially interesting on denser cell populations where cell clusters and touching cells are more the rule than the exception. Wouldn't most of these clusters be excluded from the analysis? And would that not introduce a biologically relevant bias towards isolated cells as compared to cells in contact with other? Also for the co-culture of cells of different size, how strict do the rules have to be to be able to detect all cells of both types without false positives or negatives?

Response: We thank the reviewer for this comment and raising the very interesting question.

We have added new results (**Supplementary Figure 31 and 32**) from the MSI Parser-based data extraction and analysis into the Supporting Information. Using the same data set from the MSI of PANC1 and PSC cells, we compare the results between the automatic workflow with MSI Parser and the manual workflow with SCiLS Lab (**Supplementary Figure 31**). These results indicate that the automated workflow could successfully recognize the heterogeneity between the PANC1 and PSC cells, which is consistent with the result from the manual extraction.

The reviewer raised a very interesting question regarding the scenario of denser cell populations. We agree that cells may behave differently at high cell density. Our strategy in the design of this automated workflow was to have it serve as a complement to our existing manual single-cell segmentation strategy, so we intentionally excluded cell clusters and focused on the single cells. We can adjust parameters to include clusters or touching cells when cell density is higher for a different purpose to prevent biological bias. Further, we agree that it is challenging to recognize the cell boundaries from cell clusters and touching cells solely based on the MS data. In response to this issue, we plan to optimize this program for more accurate extraction of cell boundaries in the overlapped cell clusters. Previous study has shown the feasibility of recognizing the cell boundary based on a combination of cell fluorescent images with the

assistance of the well-established cell segmentation platform, CellProfiler (<https://cellprofiler.org/>) (*PNAS*, 119, e2114365119 (2022)). These advances in single cell recognition have inspired a future direction for us to improve the current program in our coming work. For the coculture system, these two cell types are very different in terms of parameters including cell shape, area, length to width ratio, etc., and thus can be easily differentiated with high confidence.

Methods:

Page 29 line 524: There seems to be a typo: What do you mean by a pressure of 350 microns?

Response: We thank the reviewer for pointing out this confusing point. We have changed the “350 microns” to “0.35 Torr”. Microns, microns of mercury, one micron is equal to 10⁻³ Torr. Microns are typically used to measure vacuums in the range of 10⁻³ to 1 Torr.

Page 31 line 561: It remains unclear where the “old” manual and where the new automated work-flows were used. A comparison of both work-flows on the same data set would be interesting.

Response: We thank the reviewer for this comment. We have added new results (**Supplementary Figure 31 and Supplementary Figure 32**) from the MSI Parser-based data extraction and analysis into the Supporting Information. Using the same data set from the MSI of PANC1 and PSC cells, we compared the results between the automatic workflow, MSI Parser, and the manual workflow with SCiLS Lab. The result indicates the automatic workflow could successfully recognize the heterogeneity between the PANC1 and PSC cells, which is consistent with the result from the manual extraction.

Page 31 line 561: How did you export mass spectra from ROIs? How did you perform peak picking for data base search?

Response: We thank the reviewer for this comment. For using the SCiLS Lab software, the overview mass spectra of each single cell ROI were manually exported to a csv. file from the “Object” panel. For using our MSI Parser program, exporting mass spectra from ROI was completed through use of the pyimzML Python package (<https://github.com/alexandrovteam/pyimzML/tree/master>). Using the coordinates of the pixels pertaining to the cell of interest, this package extracts and averages the spectra, providing the user with an interactive image of the averaged spectrum as well as a peak list. Our current search criteria included a resolution of 7000 and a peak intensity threshold of 2000 on peak picking. These and other parameters, such as m/z tolerance, can be set by the user within the GUI of MSI Parser. Details about exporting the mass spectra from ROI have been added into the Method Section as below.

“Region of Interests (ROIs) for the SCs were picked using the SCiLS Lab Pro manually or a custom-developed MSI Parser program by the coregistration of MSI image and the optical images.”

“A platform, termed as MSI Parser, for automated data analysis was developed to contribute high-throughput capabilities to the presented pipeline (**Supplementary Figure 30**). For data analysis, 8 m/z values corresponding to each background (no cells detected) and foreground (cells detected) were

assigned and summed. For each pixel, the background sum intensity was subtracted from the coordinating foreground sum intensity. Pixels with a sum intensity greater than or equal to six-times the average background pixel intensity were assigned as cells, denoted with an intensity value of 1, and all other pixels were assigned as background, with an intensity value of 0. Following this, a median filter was applied to eliminate non-cellular artifacts. Using the pyimzML python-based package, cells were defined, and their coordinates were extracted. Our current search criteria included a resolution of 7000 and a peak intensity threshold of 2000 on peak picking. Cells were eliminated automatically based on defined area thresholds. Algorithm accuracy was confirmed through comparison of selected cell regions of interest (ROI) with optical images. Following this validation, the spectrum for each pixel of a given cell was extracted for an m/z of interest and averaged across the area of the cell.”

Page 32 line 577: All cells for this analysis seem to be “hand picked” to clearly resemble one group or the other. What about cells on the sample that were not distinctly identified? How do results compare if cell picking is done with the new automated software tools?

Response: We thank the reviewer for this comment. We worked meticulously to include filters in our program to accommodate cells of unique morphologies, such as filtering for the number of pixels a cell occupies as opposed to filtering by cellular radius. We also intentionally designed the layout of the GUI to enable side-by-side viewing of the optical image and the segmented cell ROIs, so the user can adjust parameters according to the cell type. We have updated new data of **Supplementary Figure 31 and Supplementary Figure 32** into the Supporting Information to illustrate the performance of the automatic workflow compared to manual ROI definition. The sentence below has been added into the Method.

“In **Supplementary 31 and 32**, a comparison was presented between the performance of ROI picking via SCiLS Lab manually and the automated MSI Parser tool.”

Page 32 line 588: The information that statistical analysis does not include the IMS dimension needs to be clarified in the results and not only in the methods. The claim, that CCS data cannot be exported separately from ROIs is not correct. To my knowledge, ScisLAB API provided by Bruker allows to extract feature lists with ion mobility data from ROIs. Also independent software such as pyxis by mass analytica allows to process IMS-MS imaging data produced with timsTOF flex instruments.

Response: We appreciate the reviewer’s suggestion about inputting the ion mobility data for the statistical analysis. We removed the previously claim about “that CCS data cannot be exported separately from ROIs”. We thank the reviewer for the great suggestion in the process of IMS-MS imaging data. Using the SCiLS lab API, the CCS data were successfully extracted, and the result has been added into Supporting Information as **Supplementary Figure 20**, please refer to the previous Response on this issue.